# Interactive Grounded Language Acquisition and Generalization in a 2D World

**Haonan Yu[1], Haichao Zhang[1] & Wei Xu[1,2]**
[1]Baidu Research, Sunnyvale USA
[2]National Engineering Laboratory for Deep Learning Technology and Applications, Beijing China
{haonanyu,zhanghaichao,wei.xu}@baidu.com

## Abstract

We build a virtual agent for learning language in a 2D maze-like world. The agent sees images of the surrounding environment, listens to a virtual teacher, and takes actions to receive rewards. It interactively learns the teacher's language from scratch based on two language use cases: sentence-directed navigation and question answering. It learns simultaneously the visual representations of the world, the language, and the action control. By disentangling language grounding from other computational routines and sharing a concept detection function between language grounding and prediction, the agent reliably interpolates and extrapolates to interpret sentences that contain new word combinations or new words missing from training sentences. The new words are transferred from the answers of language prediction. Such a language ability is trained and evaluated on a population of over 1.6 million distinct sentences consisting of 119 object words, 8 color words, 9 spatial-relation words, and 50 grammatical words. The proposed model significantly outperforms five comparison methods for interpreting zero-shot sentences. In addition, we demonstrate human-interpretable intermediate outputs of the model in the appendix.

## 1 Introduction

Some empiricists argue that language may be learned based on its usage (Tomasello, 2003). Skinner (1957) suggests that the successful use of a word reinforces the understanding of its meaning as well as the probability of it being used again in the future. Bruner (1985) emphasizes the role of social interaction in helping a child develop the language, and posits the importance of the feedback and reinforcement from the parents during the learning process. This paper takes a positive view of the above behaviorism and tries to explore some of the ideas by instantiating them in a 2D virtual world where *interactive* language acquisition happens. This interactive setting contrasts with a common learning setting in that language is learned from dynamic interactions with environments instead of from static labeled data.

Language acquisition can go beyond mapping language as input patterns to output labels for merely obtaining high rewards or accomplishing tasks. We take a step further to require the language to be *grounded* (Harnad, 1990). Specifically, we consult the paradigm of procedural semantics (Woods, 2007) which posits that words, as abstract procedures, should be able to pick out referents. We will attempt to explicitly link words to environment concepts instead of treating the whole model as a black box. Such a capability also implies that, depending on the interactions with the world, words would have particular meanings in a particular context and some content words in the usual sense might not even have meanings in our case. As a result, the goal of this paper is to acquire "in-context" word meanings regardless of their suitability in all scenarios.

On the other hand, it has been argued that a child's exposure to adult language provides inadequate evidence for language learning (Chomsky, 1991), but some induction mechanism should exist to bridge this gap (Landauer & Dumais, 1997). This property is critical for any AI system to learn an infinite number of sentences from a finite amount of training data. This type of generalization problem is specially addressed in our problem setting. After training, we want the agent to generalize to interpret *zero-shot* sentences of two types:

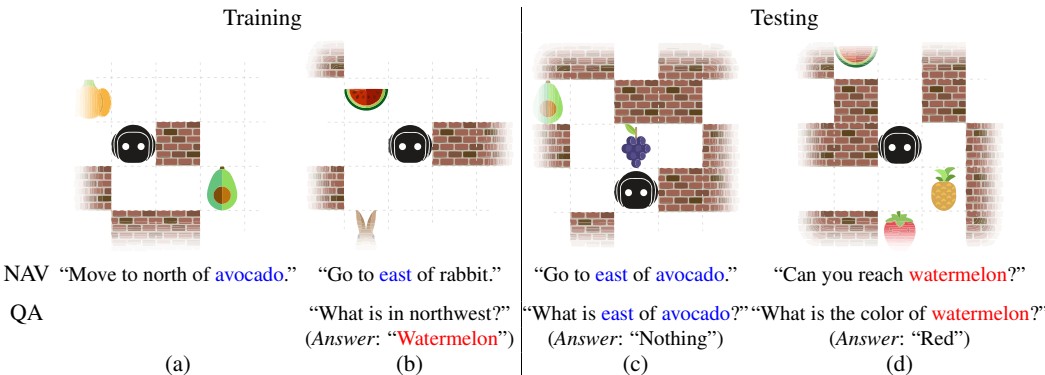

Figure 1: An illustration of XWORLD and the two language use cases. (a) and (b): A mixed training of NAV and QA. (c): Testing ZS1 sentences contain a new combination of words ("east" and "avocado") that never appear together in any training sentence. (d): Testing ZS2 sentences contain a new word ("watermelon") that never appears in any training sentence but is learned from a training answer. This figure is only a conceptual illustration of language generalization; in practice it might take many training sessions before the agent can generalize. (Due to space limitations, the maps are only partially shown.)

1) *interpolation*, new combinations of previously seen words for the same use case, or
2) *extrapolation*, new words transferred from other use cases and models.

In the following, we will call the first type ZS1 sentences and the second type ZS2 sentences. Note that so far the zero-shot problems, addressed by most recent work (Hermann et al., 2017; Chaplot et al., 2018) of interactive language learning, belong to the category of ZS1. In contrast, a reliable interpretation of ZS2 sentences, which is essentially a *transfer learning* (Pan & Yang, 2010) problem, will be a major contribution of this work.

We created a 2D maze-like world called XWORLD (Figure 1), as a testbed for interactive grounded language acquisition and generalization.[1] In this world, a virtual agent has two language use cases: navigation (NAV) and question answering (QA). For NAV, the agent needs to navigate to correct places indicated by language commands from a virtual teacher. For QA, the agent must correctly generate single-word answers to the teacher's questions. NAV tests language comprehension while QA additionally tests language prediction. They happen simultaneously: When the agent is navigating, the teacher might ask questions regarding its current interaction with the environment. Once the agent reaches the target or the time is up, the current *session* ends and a new one is randomly generated according to our configuration (Appendix B). The ZS2 sentences defined in our setting require word meanings to be transferred from single-word answers to sentences, or more precisely, *from language prediction to grounding*. This is achieved by establishing an explicit link between grounding and prediction via a common concept detection function, which constitutes the major novelty of our model. With this transferring ability, the agent is able to comprehend a question containing a new object learned from an answer, without retraining the QA pipeline. It is also able to navigate to a freshly taught object without retraining the NAV pipeline.

It is worthwhile emphasizing that this seemingly "simple" world in fact poses great challenges for language acquisition and generalization, because:

○ *The state space is huge.* Even for a $7 \times 7$ map with 15 wall blocks and 5 objects selected from 119 distinct classes, there are already octillions ($10^{27}$) of possible different configurations, not to mention the intra-class variance of object instances (see Figure 16 in the appendix). For two configurations that only differ in one block, their successful navigation paths could be completely different. This requires an accurate perception of the environment. Moreover, the configuration constantly changes from session to session, and from training to testing. In particular, the target changes across sessions in both location and appearance.

---

[1] https://github.com/PaddlePaddle/XWorld

○ *The goal space implied by the language for navigation is huge.* For a vocabulary containing only 185 words, the total number of distinct commands that can be said by the teacher conforming to our defined grammar is already over half a million. Two commands that differ by only one word could imply completely different goals. This requires an accurate grounding of language.

○ *The environment demands a strong language generalization ability from the agent.* The agent has to learn to interpret zero-shot sentences that might be as long as 13 words. It has to "plug" the meaning of a new word or word combination into a familiar sentential context while trying to still make sense of the unfamiliar whole. The recent work (Hermann et al., 2017; Chaplot et al., 2018) addresses ZS1 (for short sentences with several words) but not ZS2 sentences, which is a key difference between our learning problem and theirs.

We describe an end-to-end model for the agent to interactively acquire language from scratch and generalize to unfamiliar sentences. Here "scratch" means that the model does not hold any assumption of the language semantics or syntax. Each sentence is simply a sequence of tokens with each token being equally meaningless in the beginning of learning. This is unlike some early pioneering systems (e.g., SHRDLU (Winograd, 1972) and ABIGAIL (Siskind, 1994)) that hard-coded the syntax or semantics to link language to a simulated world–an approach that presents scalability issues. There are two aspects of the interaction: one is with the teacher (i.e., language and rewards) and the other is with the environment (e.g., stepping on objects or hitting walls). The model takes as input RGB images, sentences, and rewards. It learns simultaneously the visual representations of the world, the language, and the action control. We evaluate our model on randomly generated XWORLD maps with random agent positions, on a population of over 1.6 million distinct sentences consisting of 119 object words, 8 color words, 9 spatial-relation words, and 50 grammatical words. Detailed analysis (Appendix A) of the trained model shows that the language is grounded in such a way that the words are capable to pick out referents in the environment. We specially test the generalization ability of the agent for handling zero-shot sentences. The average NAV success rates are 84.3% for ZS1 and 85.2% for ZS2 when the zero-shot portion is half, comparable to the rate of 90.5% in a normal language setting. The average QA accuracies are 97.8% for ZS1 and 97.7% for ZS2 when the zero-shot portion is half, almost as good as the accuracy of 99.7% in a normal language setting.

## 2 MODEL

Our model incorporates two objectives. The first is to maximize the cumulative reward of NAV and the second is to minimize the classification cost of QA. For the former, we follow the standard reinforcement learning (RL) paradigm: the agent learns the action at every step from reward signals. It employs the actor-critic (AC) algorithm (Sutton & Barto, 1998) to learn the control policy (Appendix E). For the latter, we adopt the standard supervised setting of Visual QA (Antol et al., 2015): the groundtruth answers are provided by the teacher during training. The training cost is formulated as the multiclass cross entropy.

### 2.1 MOTIVATION

The model takes two streams of inputs: images and sentences. The key is how to model the language grounding problem. That is, the agent must link (either implicitly or explicitly) language concepts to environment entities to correctly take an action by understanding the instruction in the current visual context. A straightforward idea would be to encode the sentence $s$ with an RNN and encode the perceived image $e$ with a CNN, after which the two encoded representations are mixed together. Specifically, let the multimodal module be $\mathbf{M}$, the action module be $\mathbf{A}$, and the prediction module be $\mathbf{P}$, this idea can be formulated as:

$$\begin{aligned} \text{NAV:} \quad & \mathbf{A}\big(\mathbf{M}(\text{RNN}(s), \text{CNN}(e))\big) \\ \text{QA:} \quad & \mathbf{P}\big(\mathbf{M}(\text{RNN}(s), \text{CNN}(e))\big). \end{aligned} \quad (1)$$

Hermann et al. (2017); Misra et al. (2017); Chaplot et al. (2018) all employ the above paradigm. In their implementations, $\mathbf{M}$ is either vector concatenation or element-wise product. For any particular word in the sentence, fusion with the image could happen anywhere starting from $\mathbf{M}$ all the way to the end, right before a label is output. This is due to the fact that the RNN folds the string of words into a compact embedding which then goes through the subsequent blackbox computations.

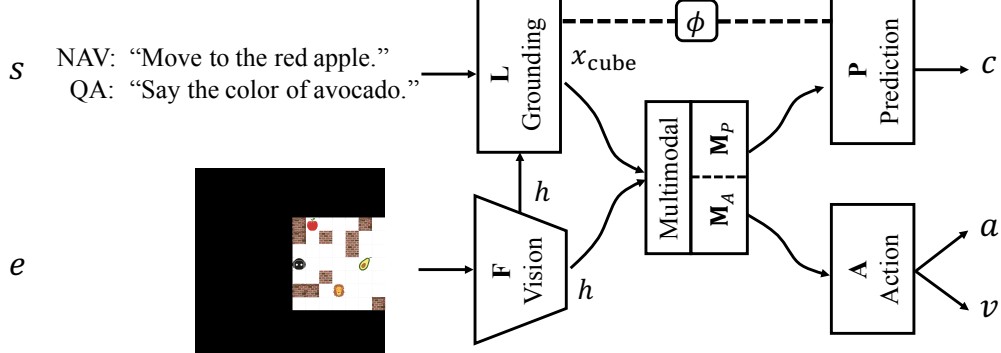

Figure 2: An overview of the model. We process $e$ by always placing the agent at the center via zero padding. This helps the agent learn navigation actions by reducing the variety of target representations. $c$, $a$, and $v$ are the predicted answer, the navigation action, and the critic value for policy gradient, respectively. $\phi$ denotes the concept detection function shared by language grounding and prediction. $\mathbf{M}_A$ generates a compact representation from $x_{\text{loc}}$ and $h$ for navigation (Appendix C).

Therefore, language grounding and other computational routines are entangled. Because of this, we say that this paradigm has an *implicit* language grounding strategy. Such a strategy poses a great challenge for processing a ZS2 sentence because it is almost impossible to predict how a new word learned from language prediction would perform in the complex entanglement involved. Thus a careful inspection of the grounding process is needed.

## 2.2 APPROACH

The main idea behind our approach is to *disentangle* language grounding from other computations in the model. This disentanglement makes it possible for us to explicitly define language grounding around a core function that is also used by language prediction. Specifically, both grounding and prediction are cast as concept detection problems, where each word (embedding) is treated as a detector. This opens up the possibility of transferring word meanings from the latter to the former. The overall architecture of our model is shown in Figure 2.

### 2.2.1 EXPLICIT GROUNDING

We begin with our definition of "grounding." We define a sentence as generally a string of words of any length. A single word is a special case of a sentence. Given a sentence $s$ and an image representation $h = \text{CNN}(e)$, we say that $s$ is *grounded* in $h$ as $x$ if

   I) $h$ consists of $M$ entities where an entity is a subset of visual features, and
   II) $x \in \{0,1\}^M$ with each entry $x[m]$ representing a binary selection of the $m$th entity of $h$. Thus $x$ is a combinatorial selection over $h$.

Furthermore, $x$ is *explicit* if

  III) it is formed by the grounding results of (some) individual words of $s$ (i.e., compositionality).

We say that a framework has an explicit grounding strategy if

  IV) *all* language-vision fusions in the framework are explicit groundings.

For our problem, we propose a new framework with an explicit grounding strategy:

$$\begin{aligned} \text{NAV:} \quad & \mathbf{A}\big(\mathbf{M}_A(x, \text{CNN}(e))\big) \\ \text{QA:} \quad & \mathbf{P}\big(\mathbf{M}_P(x, \text{CNN}(e))\big), \end{aligned} \tag{2}$$

where the sole language-vision fusion $x$ in the framework is an explicit grounding. Notice in the above how the grounding process, as a "bottleneck," allows only $x$ but not other linguistic information to flow to the downstream of the network. That is, $\mathbf{M}_A$, $\mathbf{M}_P$, $\mathbf{A}$, and $\mathbf{P}$ *all* rely on grounded

results but *not* on other sentence representations. By doing so, we expect $x$ to summarize all the necessary linguistic information for performing the tasks.

The benefits of this framework are two-fold. First, the explicit grounding strategy provides a *conceptual abstraction* (Garnelo et al., 2016) that maps high-dimensional linguistic input to a lower-dimensional conceptual state space and abstracts away irrelevant input signals. This improves the generalization for similar linguistic inputs. Given $e$, all that matters for NAV and QA is $x$. This guarantees that the agent will perform exactly in the same way on the same image $e$ even given different sentences as long as their grounding results $x$ are the same. It disentangles language grounding from subsequent computations such as obstacle detection, path planning, action making, and feature classification, which all should be inherently language-independent routines. Second, because $x$ is explicit, the roles played by the individual words of $s$ in the grounding are interpretable. This is in contrast to Eq. 1 where the roles of individual words are unclear. The interpretability provides a possibility of establishing a link between language grounding and prediction, which we will perform in the remainder of this section.

### 2.2.2 INSTANTIATION OF EXPLICIT GROUNDING

Let $h \in \mathbb{R}^{N \times D}$ be a spatially flattened feature cube (originally in 3D, now the 2D spatial domain collapsed into 1D for notational simplicity), where $D$ is the number of channels and $N$ is the number of locations in the spatial domain. We adopt three definitions for an entity:

1) a feature vector at a particular image location,
2) a particular feature map along the channel dimension, and
3) a scalar feature at the intersection of a feature vector and a feature map.

Their grounding results are denoted as $x_{\text{loc}}(s, h) \in \{0, 1\}^N$, $x_{\text{feat}}(s, h) \in \{0, 1\}^D$, and $x_{\text{cube}}(s, h) \in \{0, 1\}^{N \times D}$, respectively. In the rest of the paper, we remove $s$ and $h$ from $x_{\text{loc}}$, $x_{\text{feat}}$, and $x_{\text{cube}}$ for notational simplicity while always assuming a dependency on them. We assume that $x_{\text{cube}}$ is a low-rank matrix that can be decomposed into the two:

$$x_{\text{cube}} = x_{\text{loc}} \cdot x_{\text{feat}}{}^\mathsf{T}.$$

To make the model fully differentiable, in the following we relax the definition of grounding so that $x_{\text{loc}} \in [0, 1]^N$, $x_{\text{feat}} \in [0, 1]^D$, and $x_{\text{cube}} \in [0, 1]^{N \times D}$. The attention map $x_{\text{loc}}$ is responsible for image spatial attention. The channel mask $x_{\text{feat}}$ is responsible for selecting image feature maps, and is assumed to be independent of the specific $h$, namely, $x_{\text{feat}}(s, h) = x_{\text{feat}}(s)$. Intuitively, $h$ can be modulated by $x_{\text{feat}}$ before being sent to downstream processings. A recent paper by de Vries et al. (2017) proposes an even earlier modulation of the visual processing by directly conditioning some of the parameters of a CNN on the linguistic input.

Finally, we emphasize that our explicit grounding, even though instantiated as a soft attention mechanism, is different from the existing visual attention models. Some attention models such as Xu et al. (2015); de Vries et al. (2017) violate definitions III and IV. Some work (Andreas et al., 2016a;b; Lu et al., 2016) violates definition IV in a way that language is fused with vision by a multilayer perceptron (MLP) after image attention. Anderson et al. (2017) proposes a pipeline similar to ours but violates definition III in which the image spatial attention is computed from a compact question embedding output by an RNN.

### 2.2.3 CONCEPT DETECTION

With language grounding disentangled, now we relate it to language prediction. This relation is a common *concept detection* function. We assume that every word in a vocabulary, as a concept, is detectable against entities of type (1) as defined in Section 2.2.1. For a meaningful detection of spatial-relation words that are irrelevant to image content, we incorporate parametric feature maps into $h$ to learn spatial features. Assume a precomputed $x_{\text{feat}}$, the concept detection operates by sliding over the spatial domain of the feature cube $h$, which can be written as a function $\phi$:

$$\phi : h, x_{\text{feat}}, u \mapsto \chi,$$

where $\chi \in \mathbb{R}^N$ is a detection score map and $u$ is a word embedding vector. This function scores the embedding $u$ against each feature vector of $h$, modulated by $x_{\text{feat}}$ that selects which feature maps to

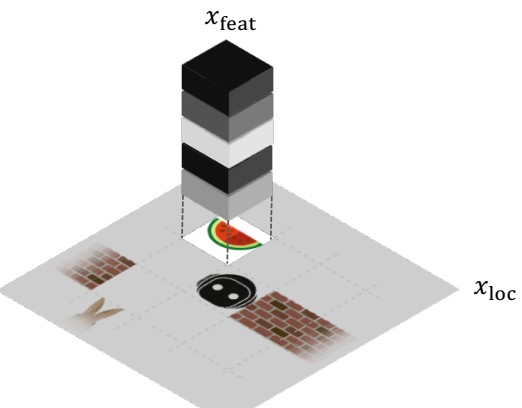

"What is the color of the object in the northeast?"

Figure 3: An illustration of the attention cube $x_{\text{cube}} = x_{\text{loc}} \cdot x_{\text{feat}}^{\mathsf{T}}$, where $x_{\text{loc}}$ attends to image regions and $x_{\text{feat}}$ selects feature maps. In this example, $x_{\text{loc}}$ is computed from "northeast." In order for the agent to correctly answer "red" (color) instead of "watermelon" (object name), $x_{\text{feat}}$ has to be computed from the sentence pattern "What ... color ...?"

use for the scoring. Intuitively, each score on $\chi$ indicates the detection response of the feature vector in that location. A higher score represents a higher detection response.

While there are many potential forms for $\phi$, we implement it as

$$\phi(h, x_{\text{feat}}, u) = h \cdot (x_{\text{feat}} \circ u), \tag{3}$$

where $\circ$ is the element-wise product. To do so, we have word embedding $u \in \mathbb{R}^D$ where $D$ is equal to the number of channels of $h$.

### 2.2.4 PREDICTION BY CONCEPT DETECTION

For prediction, we want to output a word given a question $s$ and an image $e$. Suppose that $x_{\text{loc}}$ and $x_{\text{feat}}$ are the grounding results of $s$. Based on the detection function $\phi$, $\mathbf{M}_P$ outputs a score vector $m \in \mathbb{R}^K$ over the entire lexicon, where each entry of the vector is:

$$m[k] = x_{\text{loc}}^{\mathsf{T}} \phi(h, x_{\text{feat}}, u_k) = x_{\text{loc}}^{\mathsf{T}} \chi_k, \tag{4}$$

where $u_k$ is the $k$th entry of the word embedding table. The above suggests that $m[k]$ is the result of weighting the scores on the map $\chi_k$ by $x_{\text{loc}}$. It represents the correctness of the $k$th lexical entry as the answer to the question $s$. To predict an answer

$$\mathbf{P}(m) = \underset{k}{\operatorname{argmax}} \big(\operatorname{softmax}(m)\big).$$

Note that the role of $x_{\text{feat}}$ in the prediction is to select which feature maps are relevant to the question $s$. Otherwise it would be confusing for the agent about what to predict (e.g., whether to predict a color or an object name). By using $x_{\text{feat}}$, we expect that different feature maps encode different image attributes (see an example in the caption of Figure 3). More analysis of $x_{\text{feat}}$ is performed in Appendix A.

### 2.2.5 GROUNDING BY CONCEPT DETECTION

To compute $x_{\text{cube}}$, we compute $x_{\text{loc}}$ and $x_{\text{feat}}$ separately.

We want $x_{\text{loc}}$ to be built on the detection function $\phi$. One can expect to compute a series of score maps $\chi$ of individual words and merge them into $x_{\text{loc}}$. Suppose that $s$ consists of $L$ words $\{w_l\}$ with $w_l = u_k$ being some word $k$ in the dictionary. Let $\tau(s)$ be a sequence of indices $\{l_i\}$ where $0 \leqslant l_i < L$. This sequence function $\tau$ decides which words of the sentence are selected and organized in what order. We define $x_{\text{loc}}$ as

$$\begin{aligned} x_{\text{loc}} &= \Upsilon\big(\phi(h, \mathbf{1}, w_{l_1}), \ldots, \phi(h, \mathbf{1}, w_{l_i}), \ldots, \phi(h, \mathbf{1}, w_{l_I})\big) \\ &= \Upsilon(\chi_{l_1}, \ldots, \chi_{l_i}, \ldots, \chi_{l_I}), \end{aligned} \tag{5}$$

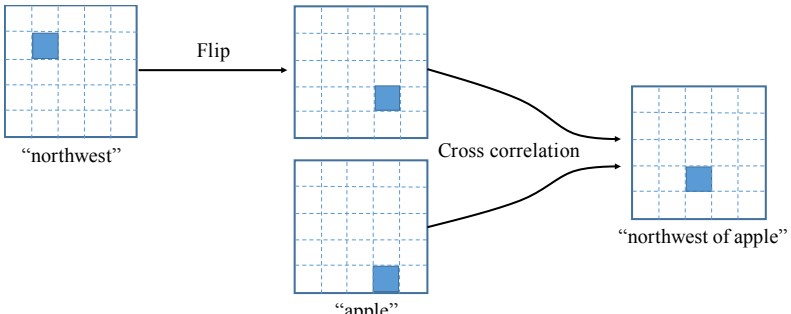

Figure 4: A symbolic example of the 2D convolution for transforming attention maps. A 2D convolution can be decomposed into two steps: flipping and cross correlation. The attention map of "northwest" is treated as an offset filter to translate that of "apple." Note that in practice, the attention is continuous and noisy, and the interpreter has to learn to find out the words (if any) to perform this convolution.

where $\mathbf{1} \in \{0,1\}^D$ is a vector of ones, meaning that it selects all the feature maps for detecting $w_{l_i}$. $\Upsilon$ is an aggregation function that combines the sequence of score maps $\chi_{l_i}$ of individual words. As such, $\phi$ makes it possible to transfer new words from Eq. 4 to Eq. 5 during test time.

If we were provided with an oracle that is able to output a parsing tree for any sentence, we could set $\tau$ and $\Upsilon$ according to the tree semantics. Neural module networks (NMNs) (Andreas et al., 2016a;b; Hu et al., 2017) rely on such a tree for language grounding. They generate a network of modules where each module corresponds to a tree node. However, labeled trees are needed for training. Below we propose to learn $\tau$ and $\Upsilon$ based on word attention (Bahdanau et al., 2015) to bypass the need for labeled structured data.

We start by feeding a sentence $s = \{w_l\}$ of length $L$ to a bidirectional RNN (Schuster & Paliwal, 1997). It outputs a compact sentence embedding $s_{\text{emb}}$ and a sequence of $L$ word context vectors $\overline{w}_l$. Each $\overline{w}_l$ summarizes the sentential pattern around that word. We then employ a meta controller called *interpreter* in an iterative manner. For the $i$th interpretation step, the interpreter computes the word attention as:

$$\tau^* \begin{cases} \text{Word attention:} & o_l^i \propto \exp\left[S_{\cos}(p^{i-1}, \overline{w}_l)\right] \\ \text{Attended context:} & \overline{w}^i = \sum_l o_l^i \overline{w}_l \\ \text{Attended word:} & s^i = \sum_l o_l^i w_l \\ \text{Interpreter state:} & p^i = \text{GRU}(p^{i-1}, \overline{w}^i) \end{cases} \tag{6}$$

where $S_{\cos}$ is cosine similarity and GRU is the gated recurrent unit (Cho et al., 2014). Here we use $\tau^*$ to represent an approximation of $\tau$ via soft word attention. We set $p^0$ to the compact sentence embedding $s_{\text{emb}}$. After this, the attended word $s^i$ is fed to the detection function $\phi$. The interpreter aggregates the score map of $s^i$ by:

$$\Upsilon \begin{cases} \text{Detection:} & y' = \text{softmax}\big(\phi(h, \mathbf{1}, s^i)\big) \\ \text{Map transform:} & x_{\text{loc}}^i = y' * y^{i-1} \\ \text{Map update gate:} & \rho^i = \sigma(W p^i + b) \\ \text{Map update:} & y^i = \rho^i x_{\text{loc}}^i + (1 - \rho^i) y^{i-1} \end{cases} \tag{7}$$

where $*$ denotes a 2D convolution, $\sigma$ is sigmoid, and $\rho^i$ is a scalar. $W$ and $b$ are parameters to be learned. Finally, the interpreter outputs $x_{\text{loc}}^I$ as $x_{\text{loc}}$, where $I$ is the predefined maximum step.

Note that in the above we formulate the map transform as a 2D convolution. This operation enables the agent to reason about spatial relations. Recall that each attention map $x_{\text{loc}}$ is egocentric. When the agent needs to attend to a region specified by a spatial relation referring to an object, it can translate the object attention with the attention map of the spatial-relation word which serves as a 2D convolutional offset filter (Figure 4). For this reason, we set $y^0$ as a one-hot map where the map

center is one, to represent the identity translation. A similar mechanism of spatial reasoning via convolution was explored by Kitaev & Klein (2017) for a voxel-grid 3D representation.

By assumption, the channel mask $x_{\text{feat}}$ is meant to be determined solely from $s$; namely, which features to use should only depend on the sentence itself, not on the value of the feature cube $h$. Thus it is computed as

$$x_{\text{feat}} = \text{MLP}\big(\text{RNN}(s)\big), \tag{8}$$

where the RNN returns an average state of processing $s$, followed by an MLP with the sigmoid activation.[2]

## 3 RELATED WORK

Our XWORLD is similar to the 2D block world MaseBase (Sukhbaatar et al., 2016). However, we emphasize the problem of grounded language acquisition and generalization, while they do not. There have been several 3D simulated worlds such as ViZDoom (Kempka et al., 2016), DeepMind Lab (Beattie et al., 2016), and Malmo (Johnson et al., 2016). Still, these other settings intended for visual perception and control, with less or no language.

Our problem setting draws inspirations from the AI roadmap delineated by Mikolov et al. (2015). Like theirs, we have a teacher in the environment that assigns tasks and rewards to the agent, potentially with a curriculum. Unlike their proposal of using only the linguistic channel, we have multiple perceptual modalities, the fusion of which is believed to be the basis of meaning (Kiela et al., 2016).

Contemporary to our work, several end-to-end systems also address language grounding problems in a simulated world with deep RL. Misra et al. (2017) maps instructions and visual observations to actions of manipulating blocks on a 2D plane. Hermann et al. (2017); Chaplot et al. (2018) learn to navigate in 3D under instructions, and both evaluate ZS1 generalization. Despite falling short of the vision challenges found in these other worlds, we have a much larger space of zero-shot sentences and additionally require ZS2 generalization, which is in fact a transfer learning (Pan & Yang, 2010) problem.

Other recent work (Andreas et al., 2017; Oh et al., 2017) on zero-shot multitask learning treats language tokens as (parsed) labels for identifying skills. In these papers, the zero-shot settings are not intended for language understanding.

The problem of grounding language in perception can perhaps be traced back to the early work of Siskind (1994; 1999), although no statistical learning was adopted at that time. Our language learning problem is related to some recent work on learning to ground language in images and videos (Yu & Siskind, 2013; Gao et al., 2016; Rohrbach et al., 2016). The navigation task is relevant to robotics navigation under language commands (Chen & Mooney, 2011; Tellex et al., 2011; Barrett et al., 2017). The question-answering task is relevant to image question answering (VQA) (Antol et al., 2015; Gao et al., 2015; Ren et al., 2015; Lu et al., 2016; Yang et al., 2016; Anderson et al., 2017; de Vries et al., 2017). The interactive setting of learning to accomplish tasks is similar to that of learning to play video games from pixels (Mnih et al., 2015).

## 4 EXPERIMENTS

We design a variety of experiments to evaluate the agent's language acquisition and generalization ability. Our model is first compared with several methods to demonstrate the challenges in XWORLD. We then evaluate the agent's language generalization ability in two different zero-shot conditions. Finally, we conclude with preliminary thoughts on how to scale our model to a 3D world.

---

[2] Note that here we drop the explicitness requirement (definition III) for Eq. 8. This choice of implementation simplicity is purely based on our current problem that requires little language compositionality when computing $x_{\text{feat}}$ (unlike $x_{\text{loc}}$). One could imagine an alternative grounding that is explicit where a single-step word attention extracts words from $s$ to compute $x_{\text{feat}}$.

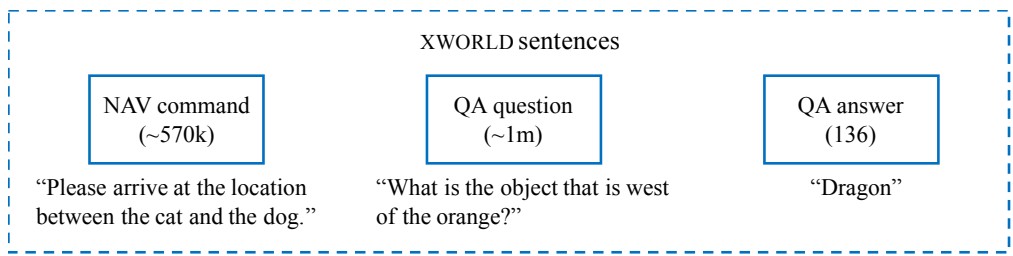

Figure 5: The three types of language data and their statistics.

## 4.1 GENERAL SETUP

For all the experiments, both the sentences and the environments change from session to session, and from training to testing. The sentences are drawn conforming to the teacher's grammar. There are three types of language data: NAV command, QA question, and QA answer, which are illustrated in Figure 5. In total, there are ∼570k NAV commands, ∼1m QA questions, and 136 QA answers (all the content words plus "nothing" and minus "between"). The environment configurations are randomly generated from octillions of possibilities of a $7 \times 7$ map, conforming to some high-level specifications such as the numbers of objects and wall blocks. For NAV, our model is evaluated on four types of navigation commands:

| | |
|---|---|
| nav_obj: | Navigate to an object. |
| nav_col_obj: | Navigate to an object with a specific color. |
| nav_nr_obj: | Navigate to a location near an object. |
| nav_bw_obj: | Navigate to a location between two objects. |

For QA, our model is evaluated on twelve types of questions (rec_* in Table 2). We refer the reader to Appendix B for a detailed description of the experiment settings.

## 4.2 COMPARISON METHODS

Four comparison methods and one ablation method are described below:

*ContextualAttention* [**CA**] A variant of our model that replaces the interpreter with a contextual attention model. This attention model uses a gated RNN to convert a sentence to a filter which is then convolved with the feature cube $h$ to obtain the attention map $x_{loc}$. The filter covers the $3 \times 3$ neighborhood of each feature vector in the spatial domain. The rest of the model is unchanged.

*StackedAttentionNet* [**SAN**] An adaptation of a model devised by Yang et al. (2016) which was originally proposed for VQA. We replace our interpreter with their stacked attention model to compute the attention map $x_{loc}$. Instead of employing a pretrained CNN as they did, we train a CNN from scratch to accommodate to XWORLD. The CNN is configured as the one employed by our model. The rest of our model is unchanged.

*VIS-LSTM* [**VL**] An adaptation of a model devised by Ren et al. (2015) which was originally proposed for VQA. We flatten $h$ and project it to the word embedding space $\mathbb{R}^D$. Then it is appended to the input sentence $s$ as the first word. The augmented sentence goes through an LSTM whose last state is used for both NAV and QA (Figure 17, Appendix D).

*ConcatEmbed* [**CE**] An adaptation of a model proposed by Mao et al. (2015) which was originally proposed for image captioning. It instantiates $\mathbf{L}$ as a vanilla LSTM which outputs a sentence embedding. Then $h$ is projected and concatenated with the embedding. The concatenated vector is used for both NAV and QA (Figure 18 Appendix D). This concatenation mechanism is also employed by Hermann et al. (2017); Misra et al. (2017).

*NavAlone* [**NAVA**] An ablation of our model that does not have the QA pipeline ($\mathbf{M}_P$ and $\mathbf{P}$) and is trained only on the NAV tasks. The rest of the model is the same.

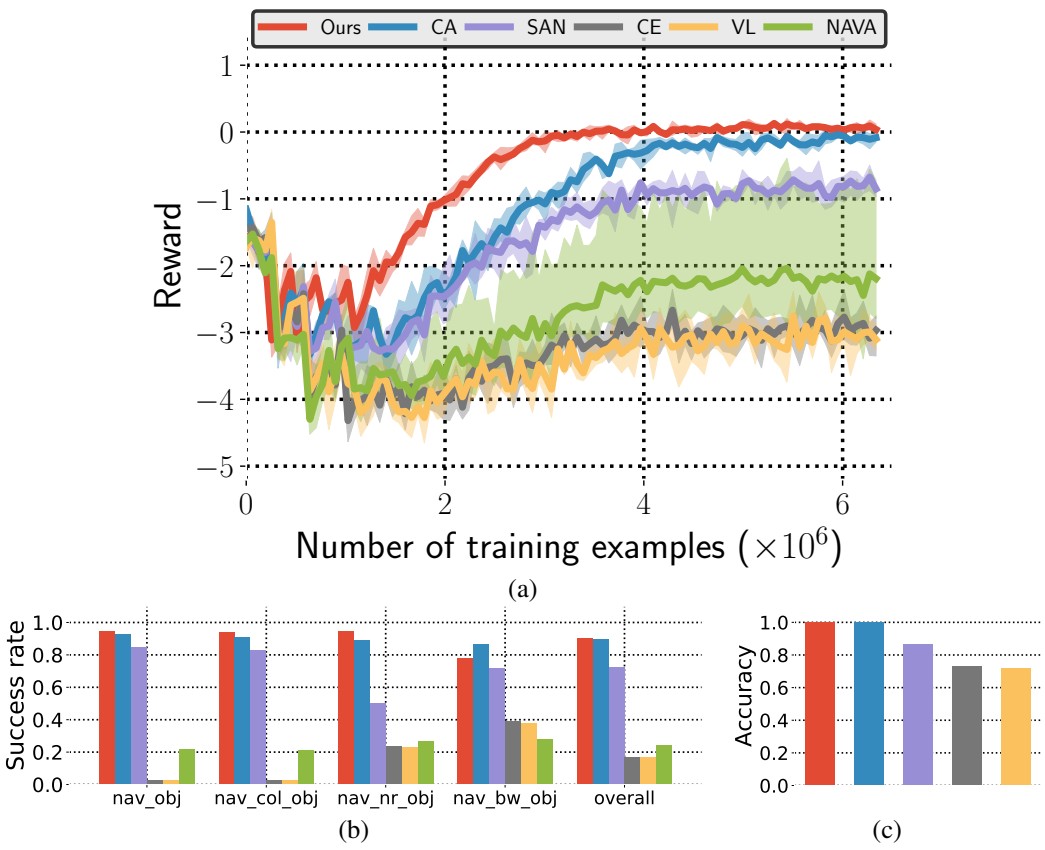

Figure 6: The basic evaluation. (a) Training reward curves. The shown reward is the accumulated discounted reward per session, averaged over every 8k training time steps. The shaded area of each curve denotes the variance among 4 random initializations of the model parameters. The reason why the curves tend to drop in the beginning is that the map difficulty increases according to our curriculum (Appendix F). (b) Navigation success rates in the test. (c) The accuracies of the answers in the test (**NAVA** is excluded because it does not train QA).

In the following experiments, we train all six approaches (four comparison methods, one ablation, and our model) with a small learning rate of $1 \times 10^{-5}$ and a batch size of 16, for a maximum of 200k minibatches. Additional training details are described in Appendix C. After training, we test each approach on 50k sessions. For NAV, we compute the average success rate of navigation where a success is defined as reaching the correct location before the time out of a session. For QA, we compute the average accuracy in answering the questions.

## 4.3 BASIC EVALUATION

In this experiment, the training and testing sentences (including NAV commands and QA questions) are sampled from the same distribution over the entire sentence space. We call it the normal language setting.[3]

The training reward curves and testing results are shown in Figure 6. **VL** and **CE** have quite low rewards without convergences. These two approaches do not use the spatial attention $x_{\text{loc}}$, and thus always attend to whole images with no focus. The region of a target location is tiny compared to the whole egocentric image (a ratio of $1 : (7 \times 2 - 1)^2 = 1 : 169$). It is possible that this

---

[3]Although some test sentences might not be seen in training (i.e., zero-shot) due to sampling, all the content words and their combinations (totaling a dozen thousands) are highly likely to be exhausted by training. Thus we consider this experiment as a normal setting, compared to the zero-shot setting in Section 4.4.

difficulty drives the models towards overfitting QA without learning useful representations for NAV. Both **CA** and **SAN** obtain rewards and success rates slightly worse than ours. This suggests that in a normal language setting, existing attention models are able to handle previously seen sentences. However, their language generalization abilities, especially on the ZS2 sentences, are usually very weak, as we will demonstrate in the next section. The ablation **NAVA** has a very large variance in its performance. Depending on the random seed, its reward can reach that of **SAN** ($-0.8$), or it can be as low as that of **CE** ($-3.0$). Comparing it to our full method, we conclude that even though the QA pipeline operates on a completely different set of sentences, it learns useful local sentential knowledge that results in an effective training of the NAV pipeline.

Note that all the four comparison methods obtained high QA accuracies (>70%, see Figure 6c), despite their distinct NAV results. This suggests that QA, as a supervised learning task, is easier than NAV as an RL task in our scenario. One reason is that the groundtruth label in QA is a much stronger training signal than the reward in NAV. Another reason might be that NAV additionally requires learning the control module, which is absent in QA.

## 4.4 LANGUAGE GENERALIZATION

Our more important question is whether the agent has the ability of interpreting zero-shot sentences. For comparison, we use **CA** and **SAN** from the previous section, as they achieved good performance in the normal language setting. For a zero-shot setting, we set up two language conditions:

*NewWordCombination* **[ZS1]** Some word pairs are excluded from both the NAV commands and the QA questions. We consider three types of unordered combinations of the content words: (*object*, *spatial relation*), (*object*, *color*), and (*object*, *object*). We randomly hold out $X\%$ of the word pairs during the training.

*NewWord* **[ZS2]** Some single words are excluded from both the NAV commands and the QA questions, but can appear in the QA answers. We randomly hold out $X\%$ of the object words during the training.

We vary the value of $X$ (12.5, 20.0, 50.0, 66.7, or 90.0) in both conditions to test how sensitive the generalization is to the amount of the held-out data. For evaluation, we report the test results only for the zero-shot sentences that contain the held-out word pairs or words. The results are shown in Figure 7.

We draw three conclusions from the results. First, the ZS1 sentences are much easier to interpret than the ZS2 sentences. Neural networks seem to inherently have this capability to some extent. This is consistent with what has been observed in some previous work (Hermann et al., 2017; Chaplot et al., 2018) that addresses the generalization on new word combinations. Second, the ZS2 sentences are difficult for **CA** and **SAN**. Even with a held-out portion as small as $X\% = 12.5\%$, their navigation success rates and QA accuracies drop up to 80% and 35%, respectively, compared to those in the normal language setting. In contrast, our model tends to maintain the same results until $X = 90.0$. Impressively, it achieves an average success rate of 60% and an average accuracy of 83% even when the number of new object words is 9 times that of seen object words in the NAV commands and QA questions, respectively! Third, in **ZS2**, if we compare the slopes of the success-rate curves with those of the accuracy curves (as shown in Figures 7e and 7f), we notice that the agent generalizes better on QA than on NAV. This further verifies our finding in the previous section that QA is in general an easier task than NAV in XWORLD. This demonstrates the necessity of evaluating NAV in addition to QA, as NAV requires additional language grounding to control.

Interestingly, we notice that `nav_bw_obj` is an outlier command type for which **CA** is much less sensitive to the increase of $X$ in **ZS2**. A deep analysis reveals that for `nav_bw_obj`, **CA** learns to cheat by looking for the image region that contains the special pattern of object pairs in the image without having to recognize the objects. This suggests that neural networks tend to exploit data in an unexpected way to achieve tasks if no constraints are imposed (Kottur et al., 2017).

To sum up, our model exhibits a strong generalization ability on both ZS1 and ZS2 sentences, the latter of which pose a great challenge for traditional language grounding models. Although we use a particular 2D world for evaluation in this work, the promising results imply the potential for scaling to an even larger vocabulary and grammar with a much larger language space.

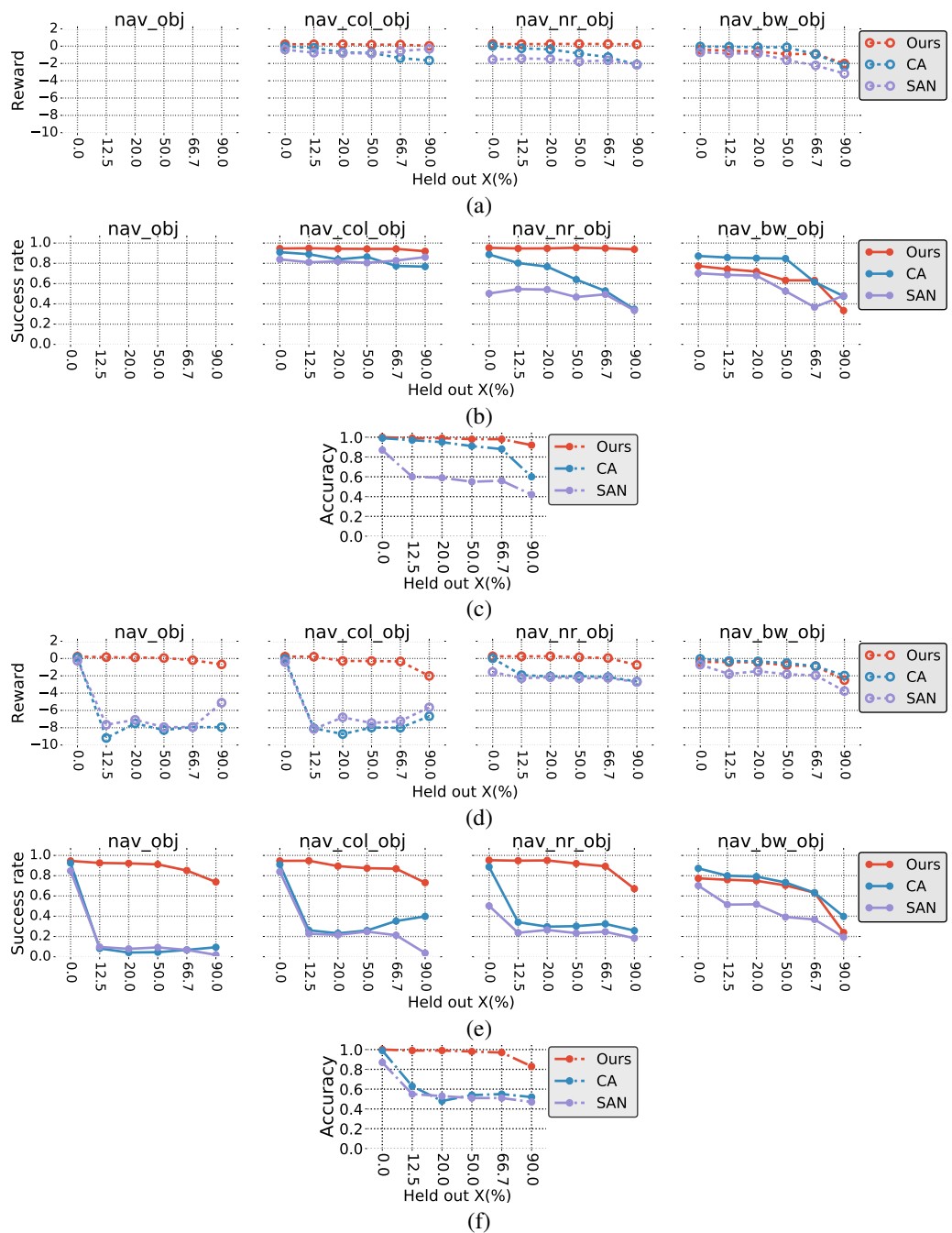

Figure 7: The test results of language generalization with a varying held-out portion of $X\%$, where $X = 0$ represents the basic evaluation in Section 4.3. (a–c) **ZS1**. (d–f) **ZS2**. For either **ZS1** or **ZS2**, from top to bottom, the three rows represent the average navigation reward per session, the average navigation success rate per session, and the average QA accuracy, respectively. (The plots of nav_obj in (a) and (b) are empty because there is no ZS1 sentence of this type by definition.)

## 4.5 HOW DOES IT ADAPT TO 3D?

We discuss the possibility of adapting our model to an agent with similar language abilities in a 3D world (e.g., Beattie et al. (2016); Johnson et al. (2016)). This is our goal for the future, but here we would like to share some preliminary thoughts. Generally speaking, a 3D world will pose a

greater challenge for vision-related computations. The key element of our model is the attention cube $x_{\text{cube}}$ that is intended for an explicit language grounding, including the channel mask $x_{\text{feat}}$ and the attention map $x_{\text{loc}}$. The channel mask only depends on the sentence, and thus is expected to work regardless of the world's dimensionality. The interpreter depends on a sequence of score maps $\chi$ which for now are computed as multiplying a word embedding with the feature cube (Eq. 3). A more sophisticated definition of $\phi$ will be needed to detect objects in a 3D environment. Additionally, the interpreter models the spatial transform of attention as a 2D convolution (Eq. 7). This assumption will be not valid for reasoning 3D spatial relations on 2D images, and a better transform method that accounts for perspective distortion is required. Lastly, the surrounding environment is only partially observable to a 3D agent. A working memory, such as an LSTM added to the action module $\mathbf{A}$, will be important for navigation in this case. Despite these changes to be made, we believe that our general explicit grounding strategy and the common detection function shared by language grounding and prediction have shed some light on the adaptation.

## 5 CONCLUSION

We have presented an end-to-end model of a virtual agent for acquiring language from a 2D world in an interactive manner, through the visual and linguistic perception channels. After learning, the agent is able to both interpolate and extrapolate to interpret zero-shot sentences that contain new word combinations or even new words. This generalization ability is supported by an explicit grounding strategy that disentangles the language grounding from the subsequent language-independent computations. It also depends on sharing a detection function between the language grounding and prediction as the core computation. This function enables the word meanings to transfer from the prediction to the grounding during the test time. Promising language acquisition and generalization results have been obtained in the 2D XWORLD. We hope that this work can shed some light on acquiring and generalizing language in a similar way in a 3D world.

### ACKNOWLEDGMENTS

We thank the anonymous reviewers for providing valuable comments and suggestions. We thank the other team members, Yuanpeng Li, Liang Zhao, Yi Yang, Zihang Dai, Qing Sun, Jianyu Wang, and Xiaochen Lian, for helpful discussions. We thank Jack Feerick and Richard Mark for proofreading. Finally, we specially thank Dhruv Batra for his feedback on an early version of this paper.

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

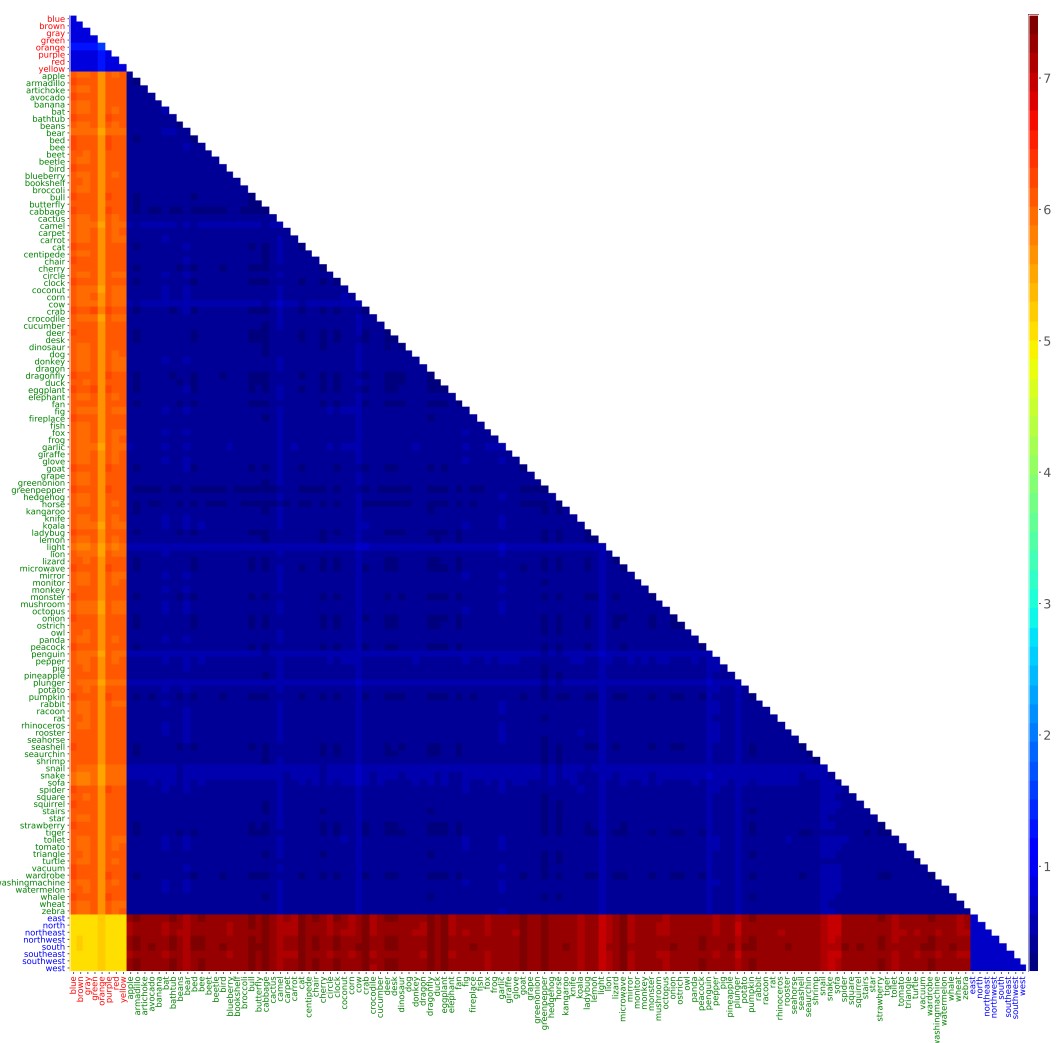

Figure 8: The Euclidean distance matrix of the 134 question groups where each group is represented by a word label. Each row (column) represents the sampled questions that have the word label as the answer. A matrix entry indicates the empirical expectation of the distance between the channel masks of the sentences from two question groups. The labels are arranged into three topics: color, object, and spatial relation. A small distance indicates that the two channel masks are similar. (Zoom in on the screen for a better view.)

## APPENDICES

## A  VISUALIZATION AND ANALYSIS

In this section, we visualize and analyze some intermediate results of our trained model.

**Channel mask** $x_{\text{feat}}$. We inspect the channel mask $x_{\text{feat}}$ which allows the model to select certain feature maps from a feature cube $h$ and predict an answer to the question $s$. We randomly sample 10k QA questions and compute $x_{\text{feat}}$ for each of them using the grounding module $\mathbf{L}$. We divide the 10k questions into 134 groups, where each group corresponds to a different answer.[4] Then we compute an Euclidean distance matrix $D$ where entry $D[i, j]$ is the average distance between the $x_{\text{feat}}$ of a question from the $i$th group and that from the $j$th group (Figure 8). It is clear that

---

[4]The word "orange" is both a color word and an object word, which is why the number of groups is one less than 119 (objects) + 8 (spatial relations without "between") + 8 (colors) = 135.

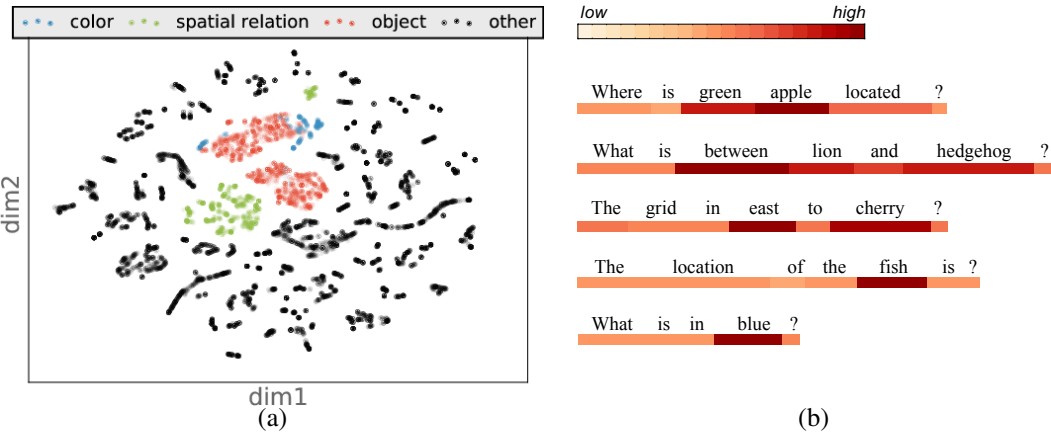

Figure 9: Visualizations of the computation of word attention. (a) Word context vectors $\overline{w}_l$. (b) The word attentions $o_l$ of several example questions. Each attention vector, represented by a color bar, shows the attention accumulated over $I$ interpretation steps.

there are three topics (object, color, and spatial relation) in the matrix. The distances computed within a topic are much smaller than those computed across topics. This demonstrates that with the channel mask, the model is able to look at different subsets of features for questions of different topics, while using the same subset of features for questions of the same topic. It also implies that the feature cube $h$ is learned to organize feature maps according to image attributes.

**Word context & attention.** To intuitively demonstrate how the interpreter works, we visualize the word context vectors $\overline{w}_l$ in Eq. 6 for a total of 20k word locations $l$ (10k from QA questions and the other 10k from NAV commands). Each word context vector is projected down to a space of 50 dimensions using PCA (Jolliffe, 1986), after which we use t-SNE (van der Maaten & Hinton, 2008; Ulyanov, 2016) to further reduce the dimensionality to 2. The t-SNE method uses a perplexity of 100 and a learning rate of 200, and runs for 1k iterations. The visualization of the 20k points is shown in Figure 9a. Recall that in Eq. 6 the word attention is computed by comparing the interpreter state $p^{i-1}$ with the word context vectors $\overline{w}_l$. In order to select the content words to generate meaningful score maps via $\phi$, the interpreter is expected to differentiate them from the remaining grammatical words based on the contexts. This expectation is verified by the above visualization in which the context vectors of the content words (in blue, green, and red) and those of the grammatical words (in black) are mostly separated. Figure 9b shows some example questions whose word attentions are computed from the word context vectors. It can be seen that the content words are successfully selected by the interpreter.

**Attention map $x_{\text{loc}}$.** Finally, we visualize the computation of the attention map $x_{\text{loc}}$. In each of the following six examples, the intermediate attention maps $x_{\text{loc}}^i$ and word attentions $o_l^i$ (in Eq. 7) of the three interpretation steps are shown from top to bottom. Each step shows the current attention map $x_{\text{loc}}^i$ overlaid on the environment image $e$. The last attention map $x_{\text{loc}}^3$ is the final output of the interpreter at the current time. Note that not all the results of the three steps are needed to generate the final output. Some results might be discarded according to the value of the update gate $\rho^i$. As a result, sometimes the interpreter may produce "bogus" intermediate attention maps which do not contribute to $x_{\text{loc}}$. Each example also visualizes the environment terrain map $x_{\text{terr}}$ (defined in Appendix C) that perfectly detects all the objects and blocks, and thus provides a good guide for the agent to navigate successfully without hitting walls or confounding targets. For a better visualization, the egocentric images are converted back to the normal view.

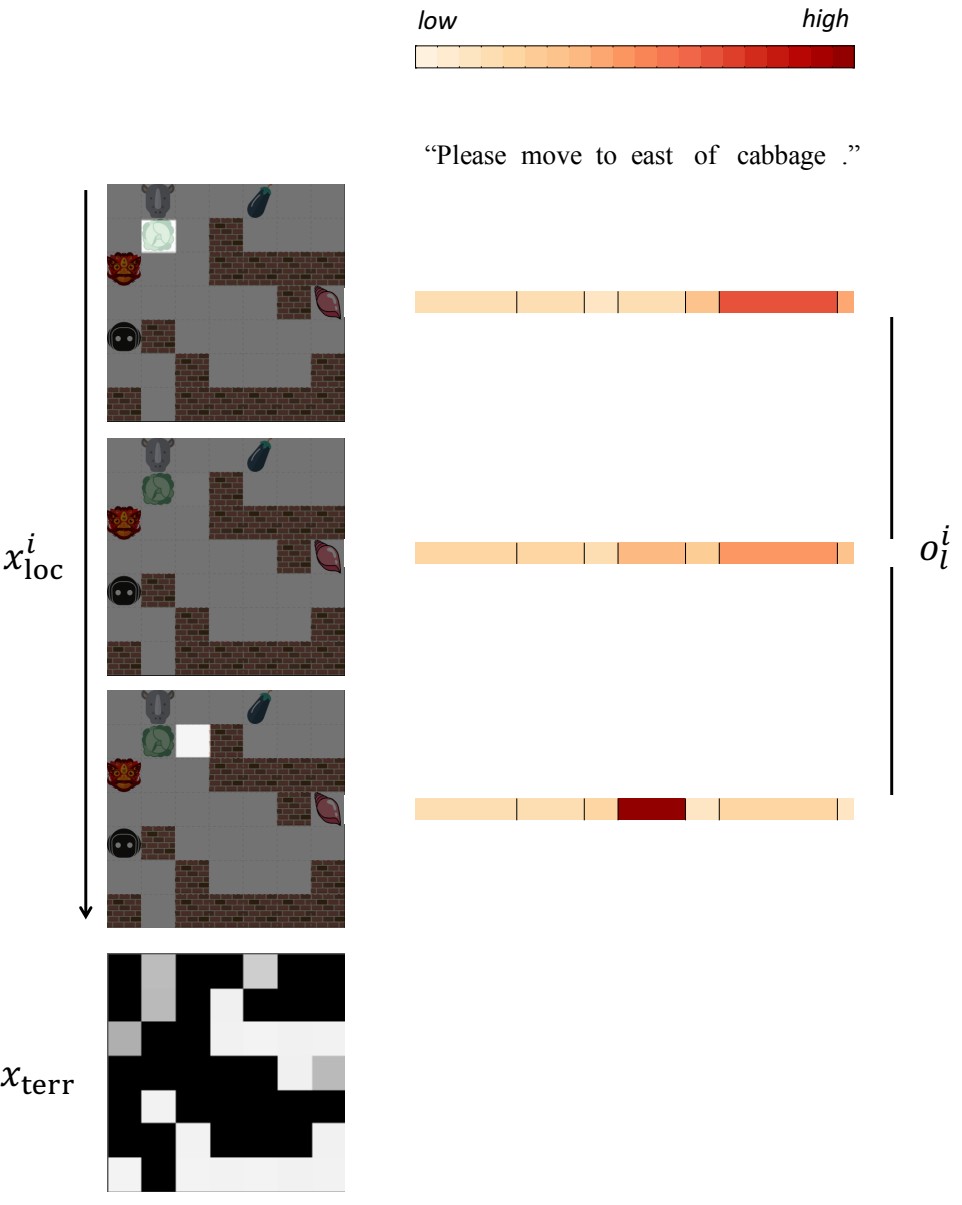

Figure 10: The first example showing how $x_{\mathrm{loc}}$ is computed.

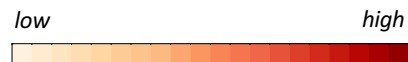

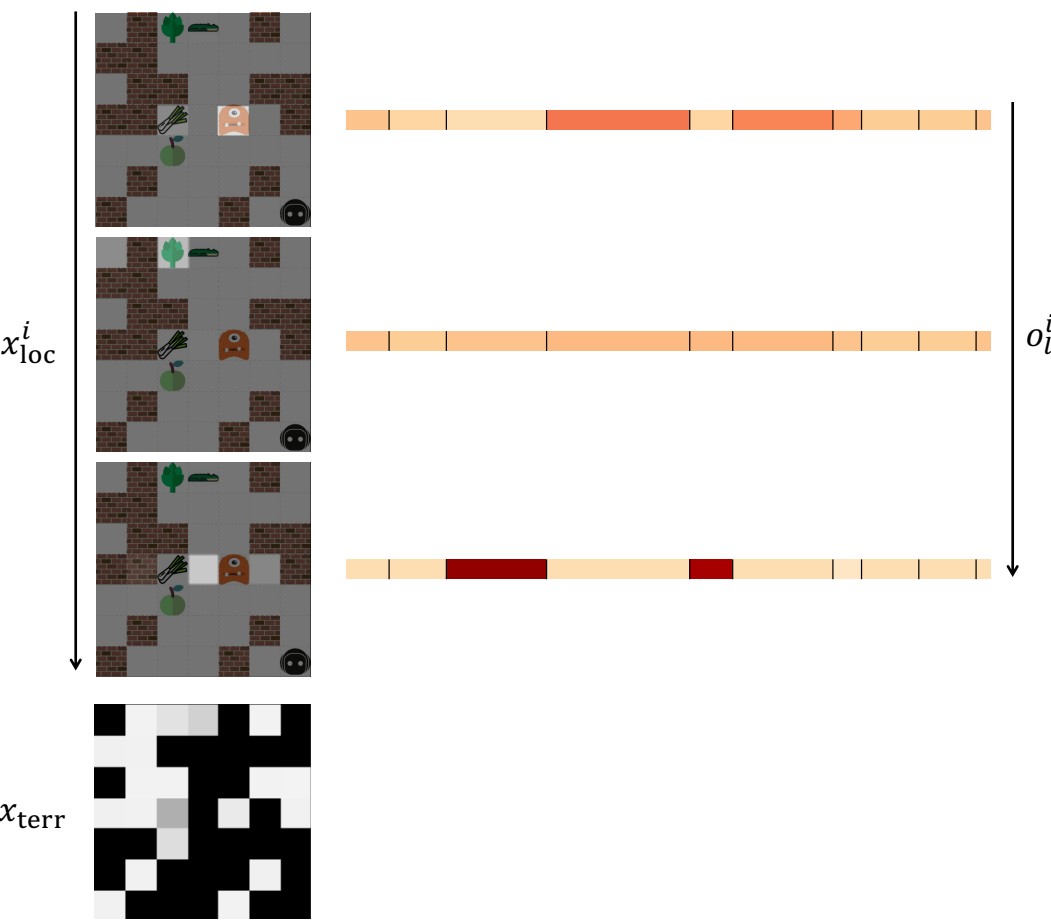

Figure 11: The second example showing how $x_{\mathrm{loc}}$ is computed.

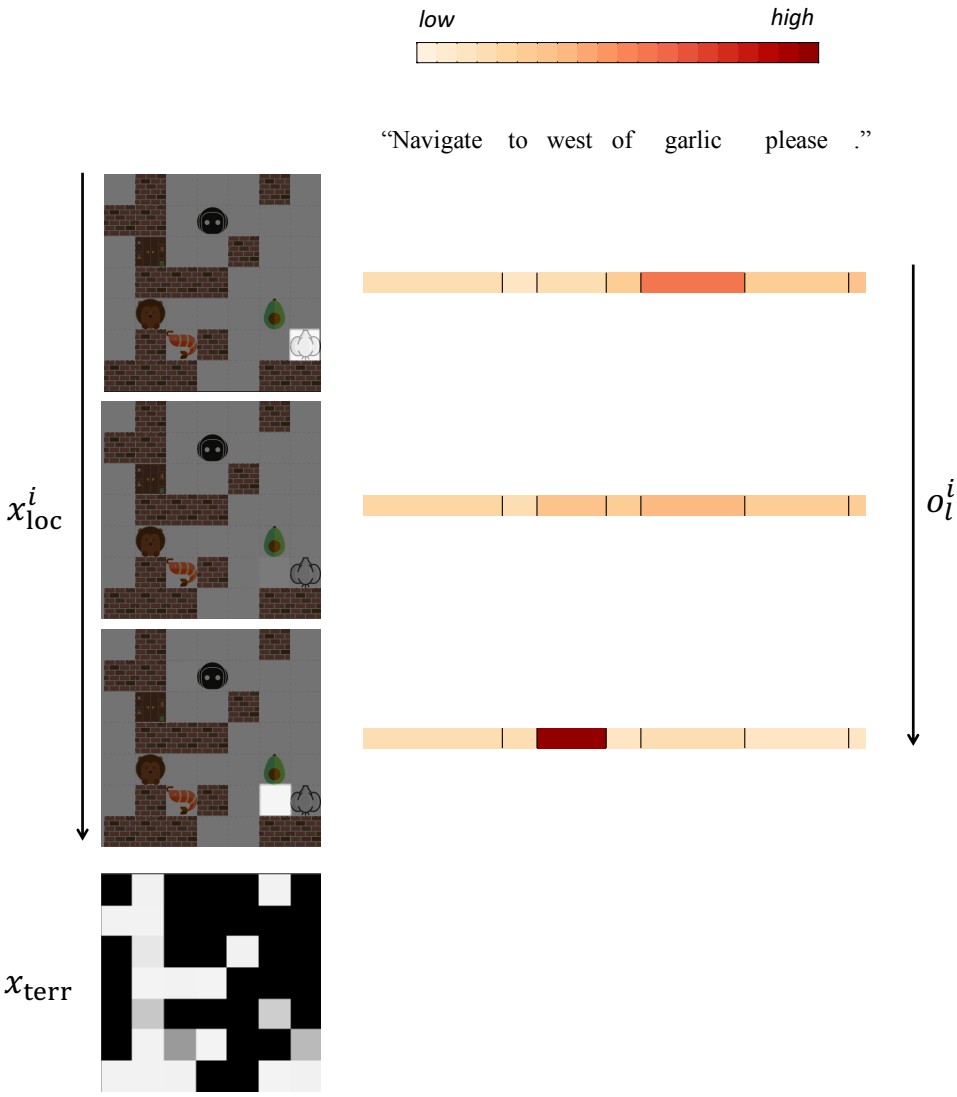

Figure 12: The third example showing how $x_{\text{loc}}$ is computed.

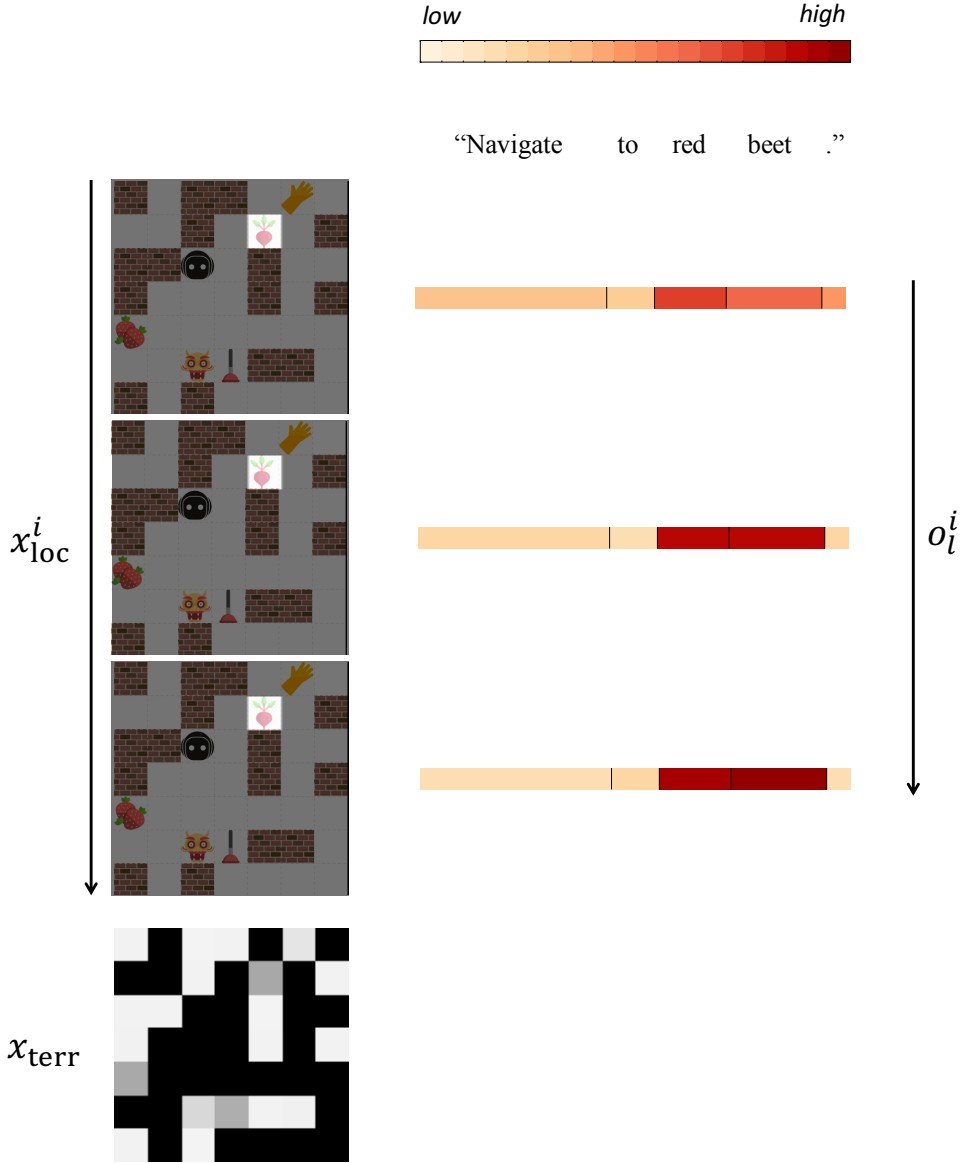

Figure 13: The fourth example showing how $x_{\mathrm{loc}}$ is computed.

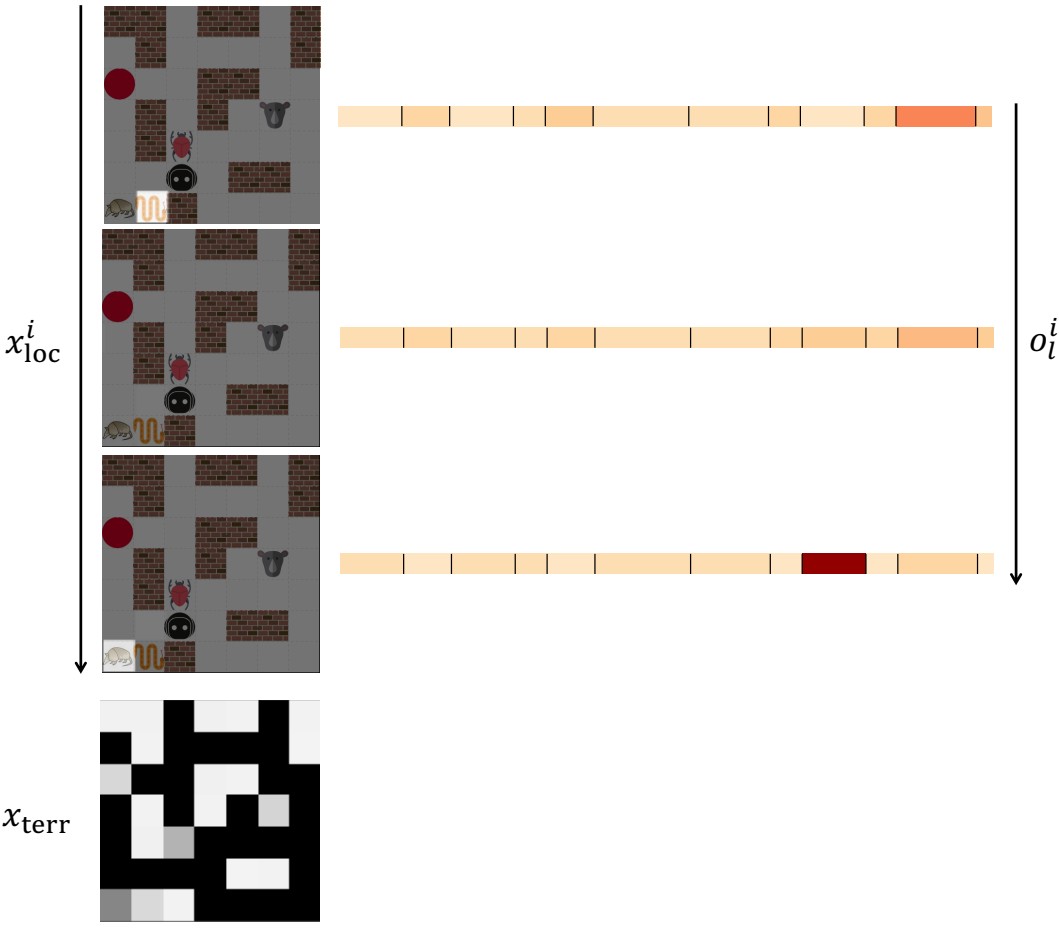

Figure 14: The fifth example showing how $x_{\text{loc}}$ is computed.

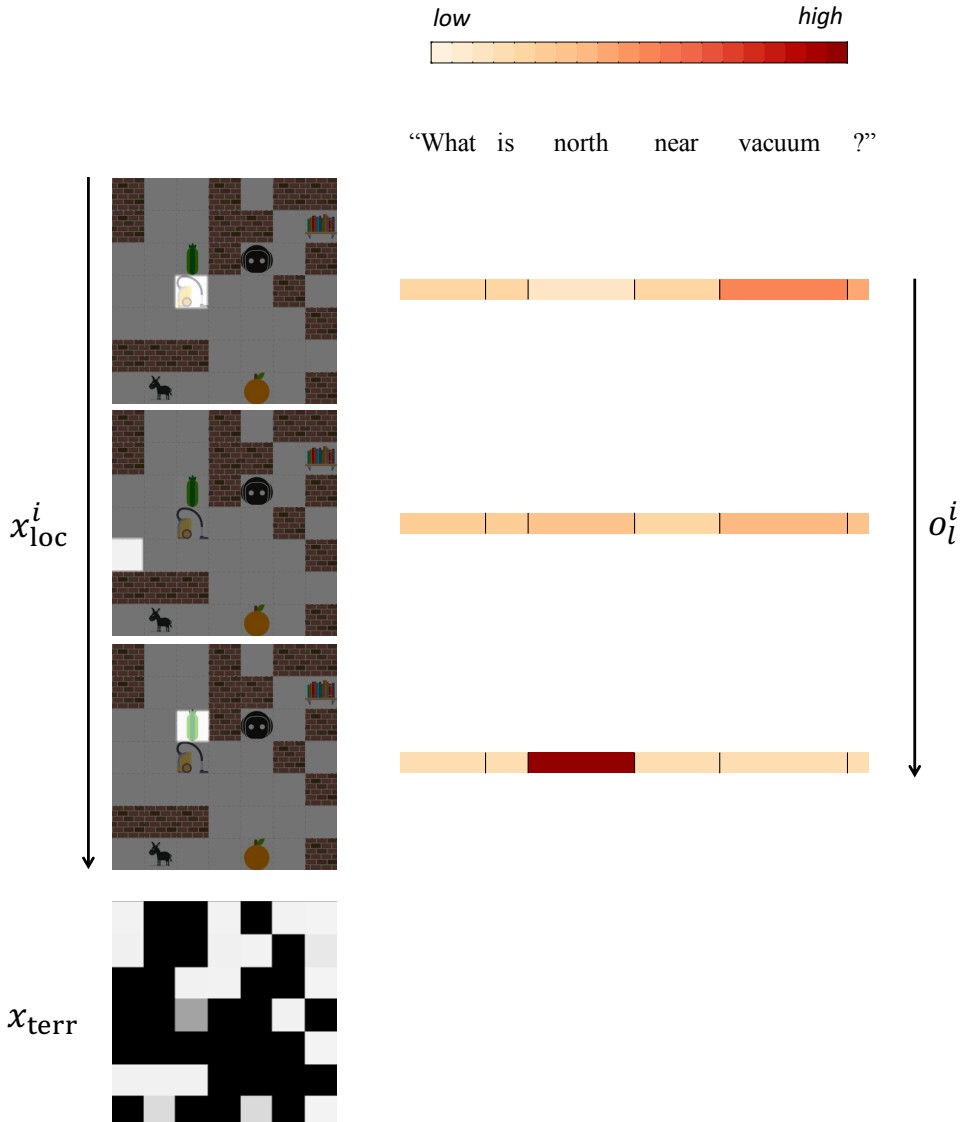

Figure 15: The sixth example showing how $x_{\text{loc}}$ is computed.

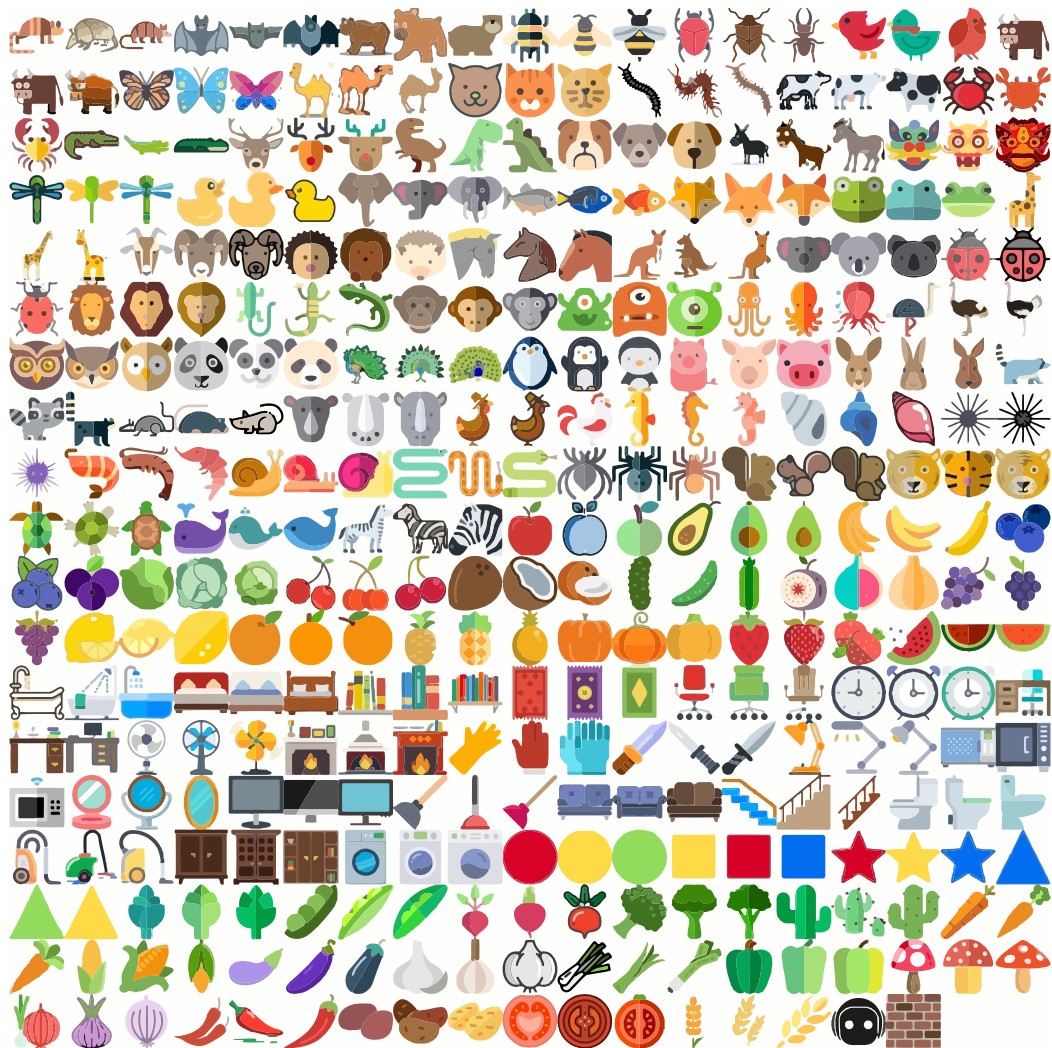

Figure 16: All the $119 \times 3 = 357$ object instances plus the agent (second-to-last) and the wall (last).

## B    XWORLD SETUP

XWORLD is configured with $7 \times 7$ grid maps. On each map, the open space for the agent has a size ranging from $3 \times 3$ to $7 \times 7$. Smaller open spaces are set up for curriculum learning (Appendix F), but not for testing. To keep the size of the environment image fixed, we pad the map with wall blocks if the open space has a size less than 7. The agent has four navigation actions in total: `left`, `right`, `up`, and `down`. In each session,

   I The maximum number of time steps is four times the map size. That is, the agent only has $7 \times 4 = 28$ steps to reach a target.

  II The number of objects on the map ranges from 1 to 5.

 III The number of wall blocks on the map ranges from 0 to 15.

 IV The positive reward when the agent reaches the correct location is $1.0$.

  V The negative rewards for hitting walls and for stepping on non-target objects are $-0.2$ and $-1.0$, respectively.

 VI The time penalty of each step is $-0.1$.

| Object | Spatial relation | Color | Other |
|---|---|---|---|
| apple, armadillo, artichoke, avocado, banana, bat, bathtub, beans, bear, bed, bee, beet, beetle, bird, blueberry, bookshelf, broccoli, bull, butterfly, cabbage, cactus, camel, carpet, carrot, cat, centipede, chair, cherry, circle, clock, coconut, corn, cow, crab, crocodile, cucumber, deer, desk, dinosaur, dog, donkey, dragon, dragonfly, duck, eggplant, elephant, fan, fig, fireplace, fish, fox, frog, garlic, giraffe, glove, goat, grape, greenonion, greenpepper, hedgehog, horse, kangaroo, knife, koala, ladybug, lemon, light, lion, lizard, microwave, mirror, monitor, monkey, monster, mushroom, octopus, onion, orange, ostrich, owl, panda, peacock, penguin, pepper, pig, pineapple, plunger, potato, pumpkin, rabbit, racoon, rat, rhinoceros, rooster, seahorse, seashell, seaurchin, shrimp, snail, snake, sofa, spider, square, squirrel, stairs, star, strawberry, tiger, toilet, tomato, triangle, turtle, vacuum, wardrobe, washingmachine, watermelon, whale, wheat, zebra | between, east, north, northeast, northwest, south, southeast, southwest, west | blue, brown, gray, green, orange, purple, red, yellow | ?, ., and, block, by, can, color, could, destination, direction, does, find, go, goal, grid, have, identify, in, is, locate, located, location, me, move, name, navigate, near, nothing, object, of, on, one, please, property, reach, say, side, target, tell, the, thing, three, to, two, what, where, which, will, you, your |

Table 1: The teacher's lexicon.

The teacher has a vocabulary size of 185. There are 9 spatial relations, 8 colors, 119 distinct object classes, and 50 grammatical words. Every object class contains 3 different instances. All object instances are shown in Figure 16. Every time the environment is reset, a number of object classes are randomly sampled and an object instance is randomly sampled for each class. There are in total 16 types of sentences that the teacher can speak, including 4 types of NAV commands and 12 types of QA questions. Each sentence type has several non-recursive templates, and corresponds to a concrete type of tasks the agent must learn to accomplish. In total there are 1,639,015 distinct sentences with 567,579 for NAV and 1,071,436 for QA. The sentence length varies between 2 and 13.

The object, spatial-relation, and color words of the teacher's language are listed in Table 1. These are the content words that can be grounded in XWORLD. All the others are grammatical words. Note that the differentiation between the content and the grammatical words is automatically learned by the agent according to the tasks. Every word is represented by an entry in the word embedding table.

The sentence types that the teacher can speak are listed in Table 2. Each type has a triggering condition about when the teacher says that type of sentences. Besides the shown conditions, an extra condition for NAV commands is that the target must be reachable from the current agent location. An extra condition for color-related questions is that the object color must be one of the eight defined colors. If at any time step there are multiple types triggered, we randomly sample one for NAV and another for QA. After a sentence type is sampled, we generate a sentence according to the corresponding sentence templates.

## C  IMPLEMENTATION DETAILS

The environment image $e$ is a $156 \times 156$ egocentric RGB image. The CNN in $\mathbf{F}$ has four convolutional layers: $(3, 3, 64), (2, 2, 64), (2, 2, 256), (1, 1, 256)$, where $(a, b, c)$ represents a layer configuration of $c$ kernels of size $a$ applied at stride width $b$. All the four layers are ReLU activated. To enable the agent to reason about spatial-relation words (e.g., "north"), we append an additional parametric cube to the convolutional output to obtain $h$. This parametric cube has the same number of channels with the CNN output, and it is initialized with a zero mean and a zero standard deviation.

The word embedding table is initialized with a zero mean and a unit standard deviation. All the gated RNNs (including the bidirectional RNN) in $\mathbf{L}$ have 128 units. All the layers in $\mathbf{L}$, unless otherwise stated, use tanh as the activation function.

| Sentence Type | Example | Triggering Condition |
|---|---|---|
| nav_obj | "Please go to the apple." | [C0] Beginning of a session. & [C1] The reference object has a unique name in the environment. |
| nav_col_obj | "Could you please move to the red apple?" | [C0] & [C2] There are multiple objects that either have the same name but different colors, or have different names but the same color. |
| nav_nr_obj | "The north of the apple is your destination." | [C0] & [C1] |
| nav_bw_obj | "Navigate to the grid between apple and banana please." | [C0] & [C3] Both reference objects have unique names in the environment and are separated by one block. |
| rec_col2obj | "What is the red object?" | [C4] There is only one object that has the specified color. |
| rec_obj2col | "What is the color of the apple?" | [C1] |
| rec_loc2obj | "Please tell the name of the object in the south." | [C5] The agent is near the reference object. |
| rec_obj2loc | "What is the location of the apple?" | [C1] & [C5] |
| rec_loc2col | "What color does the object in the east have?" | [C5] |
| rec_col2loc | "Where is the red object located?" | [C4] & [C5] |
| rec_loc_obj2obj | "Identify the object which is in the east of the apple." | [C1] & [C6] There is an object near the reference object. |
| rec_loc_obj2col | "What is the color of the east to the apple?" | [C1] & [C6] |
| rec_col_obj2loc | "Where is the red apple?" | [C2] & [C5] |
| rec_bw_obj2obj | "What is the object between apple and banana?" | [C7] Both reference objects have unique names in the environment and are separated by a block. |
| rec_bw_obj2loc | "Where is the object between apple and banana?" | [C7] & [C8] The agent is near the block which is between the two reference objects. |
| rec_bw_obj2col | "What is the color of the object between apple and banana?" | [C7] |

Table 2: All the sixteen sentence types of the teacher.

For NAV, $x_{\text{loc}}$ is used as the target to reach on the image plane. However, knowing this alone does not suffice. The agent must also be aware of walls and possibly confounding targets (other objects) in the environment. Toward this end, $\mathbf{M}_A$ further computes an environment terrain map $x_{\text{terr}} = \sigma(hf)$ where $f \in \mathbb{R}^D$ is a parameter vector to be learned and $\sigma$ is sigmoid. We expect that $x_{\text{terr}}$ detects any blocks informative for navigation. Note that $x_{\text{terr}}$ is unrelated to the specific command; it solely depends on the current environment. After stacking $x_{\text{loc}}$ and $x_{\text{terr}}$ together, $\mathbf{M}_A$ feeds them to another CNN followed by an MLP. The CNN has two convolutional layers $(3, 1, 64)$ and $(3, 1, 4)$, both with paddings of $1$. It is followed by a three-layer MLP where each layer has $512$ units and is ReLU activated.

The action module $\mathbf{A}$ contains a two-layer MLP of which the first layer has $512$ ReLU activated units and the second layer is softmax whose output dimension is equal to the number of actions.

We use adagrad (Duchi et al., 2011) with a learning rate of $10^{-5}$ for stochastic gradient descent (SGD). The reward discount factor is set to $0.99$. All the parameters have a default weight decay of $10^{-4} \times 16$. For each layer, its parameters have zero mean and a standard deviation of $1 / \sqrt{K}$, where $K$ is the number of parameters of that layer. We set the maximum interpretation step $I = 3$. The whole model is trained end to end.

# D   BASELINE DETAILS

Some additional implementation details of the baselines in Section 4.3 are described below.

**[CA]** Its RNN has $512$ units.

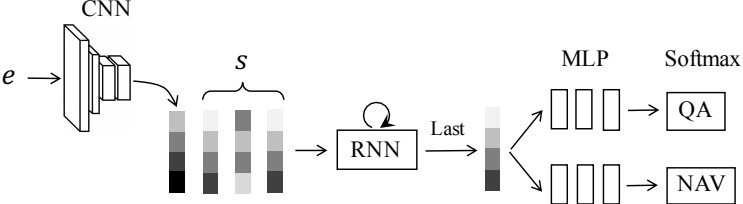

Figure 17: An overview of the baseline **VL**. The computations of NAV and QA only differ in the last MLPs.

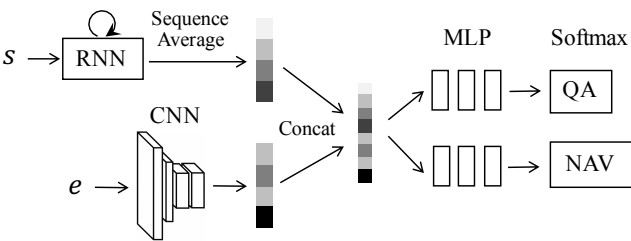

Figure 18: An overview of the baseline **CE**. The computations of NAV and QA only differ in the last MLPs.

**[VL]** Its CNN has four convolutional layers $(3, 2, 64)$, $(3, 2, 64)$, $(3, 2, 128)$, and $(3, 1, 128)$. This is followed by a fully-connected layer of size $512$, which projects the feature cube to the word embedding space. The RNN has $512$ units. For either QA or NAV, the RNN's last state goes through a three-layer MLP of which each layer has $512$ units (Figure 17).

**[CE]** It has the same layer-size configuration with **VL** (Figure 18).

**[SAN]** Its RNN has $256$ units. Following the original approach (Yang et al., 2016), we use two attention layers.

All the layers of the above baselines are ReLU activated.

## E    EXPLORATION AND EXPERIENCE REPLAY

The agent has one million exploration steps in total, and the exploration rate $\lambda$ decreases linearly from $1$ to $0.1$. At each time step, the agent takes an action $a \in \{\texttt{left}, \texttt{right}, \texttt{up}, \texttt{down}\}$ with a probability of

$$\lambda \cdot \frac{1}{4} + (1 - \lambda) \cdot \pi_\theta(a|s, e),$$

where $\pi_\theta$ is the current policy, and $s$ and $e$ denote the current command and environment image, respectively. To stabilize the learning, we also employ experience replay (ER) (Mnih et al., 2015). The environment inputs, rewards, and the actions taken by the agent in the most recent 10k time steps are stored in a replay buffer. During training, each time a minibatch $\{a_i, s_i, e_i, r_i\}_{i=1}^N$ is sampled from the buffer, using the rank-based sampler (Schaul et al., 2016) which has proven to increase the training efficiency by prioritizing rare experiences. Then we compute the gradient as:

$$-\sum_{i=0}^{N} \big( \nabla_\theta \log \pi_\theta(a_i|s_i, e_i) + \nabla_\theta v_\theta(s_i, e_i) \big) \big( r_i + \gamma v_{\theta^-}(s_i', e_i') - v_\theta(s_i, e_i) \big),$$

where $i$ is the sample index in the batch, $s_i'$ and $e_i'$ are the command and image in the next time step, $v$ is the value function, $\theta$ are the current parameters, $\theta^-$ are the target parameters that have an update delay, and $\gamma$ is the discount factor. This gradient maximizes the expected reward while minimizing the temporal-difference (TD) error. Note that because of ER, our AC method is off-policy. To avoid introducing biases into the gradient, importance ratios are needed. However, we ignored them in the above gradient for implementation simplicity. We found that the current implementation worked well in practice for our problem.

## F  CURRICULUM LEARNING

We exploit curriculum learning (Bengio et al., 2009) to gradually increase the environment complexity to help the agent learn. The following quantities are increased in proportional to $\min(1, G' / G)$, where $G'$ is the number of sessions trained so far and $G$ is the total number of curriculum sessions:

   I  The size of the open space on the environment map.

  II  The number of objects in the environment.

 III  The number of wall blocks.

 IV  The number of object classes that can be sampled from.

  V  The lengths of the NAV command and the QA question.

We found that this curriculum is important for an efficient learning. Specifically, the gradual changes of quantities IV and V are supported by the findings of Siskind (1996) that children learn new words in a linguistic corpus much faster after partial exposure to the corpus. In the experiments, we set $G = 25$k during training while do *not* have any curriculum during test (i.e., testing with the maximum difficulty).

