# OpenReview forum: "Interactive Grounded Language Acquisition and Generalization in a 2D World"
_ICLR.cc/2018/Conference — Accept (Poster)_

### Official Review · AnonReviewer2 · 2017-11-24

**Rating:** 7
**Confidence:** 4

**Review:**

This paper introduces a new task that combines elements of instruction following
and visual question answering: agents must accomplish particular tasks in an
interactive environment while providing one-word answers to questions about
features of the environment. To solve this task, the paper also presents a new
model architecture that effectively computes a low-rank attention over both
positions and feature indices in the input image. It uses this attention as a
common bottleneck for downstream predictors that select actions and answers to
questions. The paper's main claim is that this model architecture enables strong
generalization: it allows the model to succeed at the instruction following task
even when given words it has only seen in QA contexts, and vice-versa.
Experiments show that on the navigation task, the proposed approach outperforms
a variety of baselines under both a normal data condition and one requiring
strong generalization.

On the whole, I think this paper does paper does a good job of motivating the
proposed modeling decisions. The approach is likely to be useful for other
researchers working on related problems. I have a few questions about the
evaluation, but most of my comments are about presentation.

EVALUATION

Is it really the case that no results are presented for the QA task, or am I
misreading one of the charts here? Given that this paper spends a lot of time
motivating the QA task as part of the training scenario, I was surprised not to
see it evaluated.

Additionally, when I first read the paper I thought that the ZS1 experiments
featured no QA training at all. However, your response to one of the sibling
comments suggests that it's still a "mixed" training setting where the sampled
QA and NAV instances happen to cover the full space. This should be made more
clear in the paper. It would be nice to know (1) how the various models perform
at QA in both ZS1 and ZS2 settings, and (2) what the actual performance is NAV
alone (even if the results are terrible).

MODEL PRESENTATION

I found section 2 difficult to read: in particular, the overloading of \Phi
with different subscripts for different output types, the general fact that
e.g. x and \Phi_x are used interchangeably, and the large number of different
variables. My best suggestions are to drop the \Phis altogether and consider
using text subscripts rather than coming up with a new name for every variable,
but there are probably other things that will also help.

OTHER NOTES

- This paper needs serious proofreading---just in the first few pages the errors
  I noticed were "in 2D environment" (in the title!), "such capability", "this
  characteristics", "such language generalization problem", "the agent need to",
  "some early pioneering system", "commands is". I gave up on keeping track at
  this point but there are many more.

- \phi in Fig 2 should be explained by the caption.

- Here's another good paper to cite for the end of 2.2.1:
  https://arxiv.org/pdf/1707.00683.pdf.

- The mechanism in 2.2.4 feels a little like
  http://aclweb.org/anthology/D17-1015

- I don't think the content on pages 12, 13, and 14 adds much to the
  paper---consider moving these to an appendix.

---

> ### Author Response · Authors · 2017-12-12
> **Evaluation of QA in ZS1 and ZS2 added; NAV alone added**
>
> Thanks for your comments! They really help a lot.
>
> First, thanks for suggesting adding the results for QA. Originally we intended to use QA as an auxiliary task to help train NAV. We didn't think of adding results for it (although we indeed had some records showing how well different methods perform in QA during the training). In the revised paper, we have included the QA classification accuracies in the normal, ZS1 and ZS2 settings (Figure 6 c, Figure 7 c and f). We believe that this addition actually demonstrates the generalization ability of our model even better (not only in NAV but also in QA). Because now we also evaluate QA in the test, we modify all the related paragraphs across the paper to emphasize this addition.
>
> We believe that the original text already clarifies (section 4.4 when defining ZS1) that ZS1 is about excluding word pairs from both NAV commands and QA questions, but not about training NAV alone. Note that training both NAV and QA together does not necessarily imply that the sampled NAV and QA instances cover the full space. For ZS1, a subspace of sentences (containing certain word pairs) is not covered. For ZS2, a different subspace of sentences (containing certain new words) is not covered. In other words, our zero-shot setting is not achieved by turning off either NAV or QA, but instead is by excluding certain sentence patterns from the training (for both NAV and QA).
>
> As requested, we also added the performance of training NAV alone without QA in the normal language setting (Figure 6). This ablation is called NAVA in the revised experiment section. An analysis of this ablation was also added (section 4.3).
>
> Thanks for suggesting citing [de Vries et al 2017] and [Kitaev and Klein 2017]. We find that they are indeed closely related to our work. We have cited and discussed them at the end of section 2.2.1 (-> 2.2.2) and section 2.2.4 (-> 2.2.5), respectively.
>
> We have simplified the notations in section 2 to keep the presentation concise as suggested. We moved the content of pages 12, 13, and 14 to Appendix A. We went through a careful round of proofreading of the revised paper. While we are still trying to get others into the proofreading process, we have uploaded the second version of the paper to facilitate possible discussions on the OpenView.

---

> > ### Comment · AnonReviewer2 · 2017-12-12
> > **Comment**
> >
> > Thanks for the extra experiments! And you're right that I misunderstood what was held out in ZS1. This revision looks good and my score is the same.

---

### Official Review · AnonReviewer3 · 2017-11-27
**Review from AnonReviewer3**

**Rating:** 6
**Confidence:** 4

**Review:**

[Overview]
In this paper, the authors proposed a unified model for combining vision, language, and action. It is aimed at controlling an agent in a virtual environment to move to a specified location in a 2D map, and answer user's questions as well. To address this problem, the authors proposed an explicit grounding way to connect the words in a sentence and spatial regions in the images. Specifically, By this way, the model could exploit the outputs of concept detection module to perform the actions and question answering as well jointly. In the experiments, the authors compared with several previous attention methods to show the effectiveness of the proposed concept detection module and demonstrated its superiority on several configurations, including in-domain and out-of-domain cases.

[Strengths]

1. I think this paper proposed interesting tasks to combine the vision, language, and actions. As we know, in a realistic environment, all three components are necessary to complete a complex tasks which need the interactions with the physical environments. The authors should release the dataset to prompt the research in this area.

2. The authors proposed a simple method to ground the language on visual input. Specifically, the authors grounded each word in a sentence to all locations of the visual map, and then perform a simple concept detection upon it. Then, the model used this intermediate representation to guide the navigation of agent in the 2D map and visual question answering as well.

3. From the experiments, it is shown that the proposed model outperforms several baseline methods in both normal tasks and out-of-domain ones. According to the visualizations, the interpreter could generate meaningful attention map given a textual query.

[Weakness]

1. The definition of explicit grounding is a bit misleading. Though the grounding or attention is performed for each word at each location of the visual map. It is a still kind of soft-attention, except that is performed for each word in a sentence. As far as I know, this has been done in several previous works, such as: (a). Hierarchical question-image co-attention for visual question answering (https://scholar.google.com/scholar?oi=bibs&cluster=15146345852176060026&btnI=1&hl=en). Lu et al. NIPS 2016. (b). Graph-Structured Representations for Visual Question Answering. Teney et al. arXiv 2016. At most recent, we have seen some more explicit way for visual grounding like: (c). Bottom-up and top-down attention for image captioning and VQA (https://arxiv.org/abs/1707.07998). Anderson et al. arXiv 2017.

2. Since the model is aimed at grounding the language on the vision based on interactions, it is worth to show how well the final model could ground the text words to each of the visual objects. Say, show the affinity matrix between the words and the objects to indicate the correlations.

[Summary]

I think this is a good paper which integrates vision, language, and actions in a virtual environment. I would foresee more and more works will be devoted to this area, considering its close connection to our daily life. To address this problem, the authors proposed a simple model to ground words on visual signals, which prove to outperform previous methods, such as CA, SAN, etc. According to the visualization, the model could attend the right region of the image for finishing a navigation and QA task. As I said, the authors should rephrase the definition of explicit grounding, to make it clearly distinguished with the previous work I listed above. Also, the authors should definitely show the grounding attention results of words and visual signal jointly, i.e., showing them together in one figure instead of separately in Figure 9 and Figure 10.

---

> ### Author Response · Authors · 2017-12-12
> **Explicit grounding clarified**
>
> Thanks for your comments!
>
> We agree with the reviewer that our original definition of explicit
> grounding had some ambiguity. Thus we added several paragraphs to
> elaborate on this. Because then the original section 2.2.1 became so long that we divided it into two (2.2.1 and 2.2.2). Specifically, we rephrased section 2.2.1 (-> 2.2.2) by giving a detailed definition about what it means for a framework to have an explicit grounding strategy. We also discussed the similarities and differences of our grounding with the related work pointed out by the reviewer at the end of section 2.2.1 (-> 2.2.2).
>
> In summary, our explicit grounding requires two extra properties on top
> of the soft attention mechanism:
>
> 1) the grounding (image attention) of a sentence is computed based on
> the grounding results of the individual words in that sentence (i.e., compositionality);
>
> 2) in the framework, there are no other types of language-vision
> fusions besides this kind of groundings by 1)
>
> One benefit of such an explicit grounding is that Eq 2 achieves a
> language "bottleneck" for downstream predictors (as Reviewer 2 pointed
> out in the comment). This bottleneck is used for both NAV and QA. It
> implies an "independence of path" property because given the image all
> that matters for the NAV and QA tasks is the attention $x$ (Eq 2). It
> guarantees, via the model architecture, that the agent will perform
> completely in the same way on the same image even given different
> sentences as long as their $x$ are the same. Also because $x$ is
> explicit, the roles played by the individual words of $s$ in
> generating $x$ are interpretable. This is in contrast to Eq 1 where
> the roles of individual words are unclear. The interpretability
> provides a possibility of establishing a link between language
> grounding and prediction. We argue that these are crucial reasons that
> account for our strong generalization in both ZS1 and ZS2 settings.
>
> We have modified the original Figure 10 so that the image attention is
> visualized jointly with the word attention. More examples are shown
> now after the modification. Because of a limited space, we moved this
> part to Appendix A and divided it into six figures (Figure 10 - Figure 15).

---

### Official Review · AnonReviewer1 · 2017-12-04
**interesting contribution**

**Rating:** 6
**Confidence:** 4

**Review:**

The paper introduces XWORLD, a 2D virtual environment with which an agent can constantly interact via navigation commands and question answering tasks. Agents working in this setting therefore, learn the language of the "teacher" and efficiently ground words to their respective concepts in the environment. The work also propose a neat model motivated by the environment and outperform various baselines.

Further, the paper evaluates the language acquisition aspect via two zero-shot learning tasks -- ZS1) A setting consisting of previously seen concepts in unseen configurations ZS2) Contains new words that did not appear in the training phase.

The robustness to navigation commands in Section 4.5 is very forced and incorrect -- randomly inserting unseen words at crucial points might lead to totally different original navigation commands right? As the paper says, a difference of one word can lead to completely different goals and so, the noise robustness experiments seem to test for the biases learned by the agent in some sense (which is not desirable). Is there any justification for why this method of injecting noise was chosen ? Is it possible to use hard negatives as noisy / trick commands and evaluate against them for robustness ?

Overall, I think the paper proposes an interesting environment and task that is of interest to the community in general. The modes and its evaluation are relevant and intuitions can be made use for evaluating other similar tasks (in 3D, say).

---

> ### Author Response · Authors · 2017-12-12
> **Robustness was for a test in an uncontrollable setting; removed in case of possible confusion**
>
> Thanks for your comments.
>
> The experiment of robustness is aimed at testing the agent in a
> scenario out of our control, such as executing navigation commands
> elicited by human after the training is done. In such case, we simply
> assume that the evaluator does not have any knowledge of the training
> process. A natural-language sentence elicited by the human evaluator
> might convey a meaning that is similar or same to a sentence generated
> by our grammar, however, it might not be that well-formed (e.g.,
> containing extra irrelevant words). One simple way of simulating this
> scenario (incompletely) is to insert noisy word embeddings into the original sentence.
>
> This preliminary experiment serves to provide some numbers to let the
> readers have a rough idea about how well the agent will perform in an
> uncontrollable setting. However, because of its minor significance and
> a possible misunderstanding, we have removed this section (4.5) from
> the original paper.

---

> > ### Comment · AnonReviewer1 · 2018-01-09
> > **Reply to revisions**
> >
> > Thank you for the reply. I have looked at the revised draft and stick to my current assessment.

---

### Public Comment · (anonymous) · 2017-11-13
**Questions about your RL setup**

Nice work!

I have a few questions about the technical details of your reinforcement learning experiments:

- I did not find the formula for the module A of the model which is mentioned in Equation (1). Is it contained in the paper?
- According to Section 3 you are using an actor-critic method. Meanwhile, your Appendix D says that you are using "Experience Replay". Can you please provide more details on what specific RL approach was used?
- According to Appendix D you train NAV and VQA pathways in parallel. Did you try training NAV only? Does you RL approach to grounding work in the absence of the additional signal from VQA?

---

> ### Author Response · Authors · 2017-11-13
> **Thanks for your comments! They are really good questions and helpful.**
>
> 1. The formula of the module A was not contained in the paper,
> primarily due to the cut down of pages. It is just a feedforward
> sub-network that approximates the value function and generates the action
> distribution, given the representation q (Fig. 2). However, as you
> have asked, we now think it might be a good idea to add it back in the
> paper.
>
> 2. Continuing on your first question, our RL method is simply
> combining AC and Experience Replay (ER). ER is mainly used to
> stabilize AC while maintaining sample efficient. This might result in
> some conflict between off-policy and on-policy, since the experiences
> sampled from the replay buffer were not generated by the current
> policy. However, we find that this works well in practice (perhaps
> because of the small replay buffer). Similar work was also proposed
> recently:
>
> 	Sample Efficient Actor-Critic With Experience Replay, Wang et al, ICLR 2017.
>
> which is more sophisticated compared to ours.
>
> More specifically, for every minibatch sampled from the replay buffer,
> we have the following gradient:
>
> -\sum_{k=0}^K(\nabla_{\theta}\log\pi_{\theta}(a_k|x_k)+\nabla_{\theta}v_{\theta}(x_k))(r+\gamma
>  v_{\theta'}(x_k')-v_{\theta}(x_k))
>
> where k is the sample index in the batch, (x_k, a_k, x_k') is the
> sampled transition, v is the value function to learn, \theta is the
> current parameters, \theta' is the target parameters that have update
> delay as in ER, and \gamma is the discount factor. This gradient
> maximizes the expected reward while minimizes the TD error.
>
> 3. We did try only training NAV without QA. However, it only worked to
> some extent for small-size maps like 3x3 or 5x5, with a much smaller
> amount of object classes. It was difficult to converge on the current
> setting of 7x7 maps with 119 object classes. Thus QA is important for
> the learning. It is not uncommon to see some auxiliary tasks used for
> a better convergence in RL problems, for example, language prediction
> and some other cost functions were used in parallel with RL:
>
> 	Grounded language learning in a simulated 3d world, Hermann et al,
> 	arxiv 1706.06551, 2017
>
> Note that QA only helps understanding of questions. The NAV commands
> and action control still need to be learned from RL. Questions and
> commands are disjoint sets of sentences. The only common part is some
> local word or phrase patterns. More importantly, QA offers the
> opportunity to assess the transferring ability of the model across
> tasks, denoted as ZS2 in the paper.
>
> We believe that such task of jointly learning language and vision is
> challenging. Even for children, it is very likely that they learn from
> a mixture of signals of the environment instead from a single task.

---

> ### Author Response · Authors · 2017-12-12
> **RL details added in the revised paper**
>
> Hi, we have updated our paper and added the details about our RL approach in Appendix E. Also, as R2 requested, we added the experiment results for training NAV alone. Please take a look at the revision if interested. Thank you!

---

### Author Response · Authors · 2018-01-03
**Versions**

For AC, reviewers, and others: to check the revisions, please compare the original version (modified: 27 Oct 2017, 13:14) and the latest version (modified: 02 Jan 2018, 10:46). Most changes were made according to the reviewers' comments. Some minor changes were made to improve the presentation.

---

### Decision · Program_Chairs · 2018-01-29
**ICLR 2018 Conference Acceptance Decision**

**Decision:**

Accept (Poster)

**Comment:**

This manuscript was reviewed by 3 expert reviewers and their evaluation is generally positive. The authors have responded to the questions asked and the reviewers are satisfied with the responses. Although the 2D environments are underwhelming (compared to 3D environments such as SUNCG, Doom, Thor, etc), one thing that distinguishes this paper from other concurrent submissions on the similar topics is the demonstration that "words learned only from a VQA-style supervision condition can be successfully interpreted in an instruction-following setting."